# RFWAVE: MULTI-BAND RECTIFIED FLOW FOR AUDIO WAVEFORM RECONSTRUCTION

**Peng Liu**
TEX AI, Transsion
`peng.liu6@transsion.com`

**Dongyang Dai**
Individual Researcher
`accum.dai@gmail.com`

**Zhiyong Wu**
Shenzhen International Graduate School, Tsinghua University
`zywu@sz.tsinghua.edu.cn`

## ABSTRACT

Recent advancements in generative modeling have significantly enhanced the reconstruction of audio waveforms from various representations. While diffusion models are adept at this task, they are hindered by latency issues due to their operation at the individual sample point level and the need for numerous sampling steps. In this study, we introduce RFWave, a cutting-edge multi-band Rectified Flow approach designed to reconstruct high-fidelity audio waveforms from Mel-spectrograms or discrete acoustic tokens. RFWave uniquely generates complex spectrograms and operates at the frame level, processing all subbands simultaneously to boost efficiency. Leveraging Rectified Flow, which targets a straight transport trajectory, RFWave achieves reconstruction with just 10 sampling steps. Our empirical evaluations show that RFWave not only provides outstanding reconstruction quality but also offers vastly superior computational efficiency, enabling audio generation at speeds up to 160 times faster than real-time on a GPU. Both an online demonstration and the source code are accessible[1].

## 1 INTRODUCTION

Audio waveform reconstruction significantly enhances the digital interactions by enabling realistic voice and sound generation for diverse applications. This technology transforms low-dimensional features, derived from raw audio data, into perceptible sounds, improving the audio experience on various platforms such as virtual assistants and entertainment systems. Autoregressive models and Generative Adversarial Networks (GANs) (Goodfellow et al., 2014) have been applied to this task, greatly advancing audio quality beyond traditional signal processing methods (Kawahara et al., 1999; Morise et al., 2016). Autoregressive methods, while effective, are hindered by slow generation speeds due to their sequential prediction of sample points (Oord et al., 2016; Kalchbrenner et al., 2018; Valin & Skoglund, 2019). In contrast, GANs predict sample points in parallel, resulting in faster generation speeds and maintaining high-quality output (Kumar et al., 2019; Yamamoto et al., 2020; Kong et al., 2020a; Siuzdak, 2023; Du et al., 2023). Consequently, GAN-based methods deliver impressive performance and are extensively utilized in real-world audio generation applications.

Despite the advancements, GAN-based waveform reconstruction models face challenges such as the necessity for complex discriminator designs and issues like instability or mode collapse (Thanh-Tung et al., 2018). In response, diffusion models for waveform reconstruction have been explored, offering stability during training and the ability to reconstruct high-quality waveforms (Chen et al., 2020; Kong et al., 2020b; gil Lee et al., 2022; Nguyen et al., 2024; Huang et al., 2022; Koizumi et al., 2022). However, these models are at least an order of magnitude slower compared to GANs. The slow generation speed in these diffusion-based waveform reconstruction models is primarily due to two factors: (1) the requirement of numerous sampling steps to achieve high-quality samples, and (2) the operation at the waveform sample point level. The latter often involves multiple upsampling

---

[1]Demo: `https://rfwave-demo.github.io/rfwave`; Code: `https://github.com/bfs18/rfwave`

operations to transition from frame rate resolution to sample rate resolution, increasing the sequence length and consequently leading to higher GPU memory usage and computational demands.

In this paper, we propose RFWave, a diffusion-type waveform reconstruction method designed to match the speed of GAN-based methods while maintaining the training stability and high sample quality of diffusion models. To overcome the challenge of slow sampling, we employ Rectified Flow (Liu et al., 2023; Lipman et al., 2023; Albergo & Vanden-Eijnden, 2023), which connects data and noise along a straight line, thereby enhancing sampling efficiency. To address the GPU memory and computational demands of sample point-level modeling, our model operates at the level of Short-Time Fourier Transform (STFT) frames, enabling more efficient processing and reducing GPU memory usage. RFWave with only 10 sampling steps can generate high-quality audio, achieving an inference speed of up to 160 times real-time on an NVIDIA GeForce RTX 4090 GPU. Additionally, the incorporation of three enhanced loss functions and an optimized sampling strategy further elevates the overall quality of the reconstructed waveforms. To our knowledge, RFWave stands as the fastest diffusion-based audio waveform reconstruction model, and it delivers superior audio quality. Furthermore, it can reconstruct waveforms from Mel-spectrograms and discrete acoustic tokens, enhancing its versatility and applicability in various audio generation tasks. Our main contributions are as follows:

1. By integrating the Rectified Flow and 3 enhanced loss functions – energy-balanced loss, overlap loss, and STFT loss – our model can reconstruct high-quality waveforms with a drastically reduced number of sampling steps.

2. We utilize a multi-band strategy, coupled with the high-efficiency ConvNeXtV2 (Woo et al., 2023) backbone, to generate different subbands concurrently. This not only assures audio quality by circumventing cumulative errors, but also boosts the synthesis speed.

3. Our model operates at the level of STFT frames, not individual waveform sample points. This approach significantly accelerates processing and reduces GPU memory usage.

4. We propose a technique for selecting sampling time points based on the straightness of the Rectified Flow transport trajectories, which enhances sample quality for free.

## 2 BACKGROUND

In this section, we describe the basic formulation of the Rectified Flow and present related works, while also detailing our correlation and distinction.

**Rectified Flow** Rectified Flow (Liu et al., 2023) presents an innovative Ordinary Differential Equation (ODE)-based framework for generative modeling and domain transfer. It introduces a method to learn a mapping that connects two distributions, $\pi_0$ and $\pi_1$ on $\mathbb{R}^d$, based on empirical observations:

$$\frac{\mathrm{d}Z_t}{\mathrm{d}t} = v(Z_t, t), \quad \text{initialized from } Z_0 \sim \pi_0, \text{ such that } Z_1 \sim \pi_1, \tag{1}$$

where $v \colon \mathbb{R}^d \times [0,1] \to \mathbb{R}^d$ represents a velocity field. The learning of this field involves minimizing a mean square objective function,

$$\min_v \mathbb{E}_{(X_0, X_1) \sim \gamma} \left[ \int_0^1 || \frac{\mathrm{d}}{\mathrm{d}t} X_t - v(X_t, t) ||^2 \, \mathrm{d}t \right], \quad \text{with} \quad X_t = \phi(X_0, X_1, t), \tag{2}$$

where $X_t = \phi(X_0, X_1, t)$ represents a time-differentiable interpolation between $X_0$ and $X_1$, with $\frac{\mathrm{d}}{\mathrm{d}t} X_t = \partial_t \phi(X_0, X_1, t)$. The $\gamma$ represents any coupling of $(\pi_0, \pi_1)$. An illustrative instance of $\gamma$ is the independent coupling $\gamma = \pi_0 \times \pi_1$, which allows for empirical sampling based on separately observed data from $\pi_0$ and $\pi_1$. Liu et al. (2023) recommended a simple choice of

$$X_t = (1-t)X_0 + tX_1 \implies \frac{\mathrm{d}}{\mathrm{d}t} X_t = X_1 - X_0. \tag{3}$$

This simplification results in linear trajectories, which are critical for accelerating the inference process. Typically, the velocity field $v$ is represented using a deep neural network. The solution to (2) is approximated through stochastic gradient methods. To approximate the ODE presented in (1),

numerical solvers are commonly employed. A prevalent technique is the forward Euler method. This approach computes values using the formula

$$Z_{t+\frac{1}{n}} = Z_t + \frac{1}{n}v(Z_t, t), \qquad \forall t \in \{0, \dots, n-1\}/n, \tag{4}$$

where the simulation is executed with a step interval of $\epsilon = 1/n$ over $n$ steps.

The velocity field has the capacity to incorporate conditional information, which is particularly essential in applications such as text-to-image generation and waveform reconstruction from compressed acoustic representation. Consequently, in such contexts, $v(Z_t, t)$ in (2) is modified to $v(Z_t, t \mid \mathcal{C})$, where $\mathcal{C}$ represents the conditional information pertinent to the corresponding $X_1$.

**Diffusion Models for Audio Waveform Reconstruction**  Diffusion models (Song et al., 2020; Ho et al., 2020) have become the de-facto choice for high-quality generation in the realm of generative models. Diffwave (Kong et al., 2020b) and WaveGrad (Chen et al., 2020) were pioneering efforts to reconstruct waveforms using diffusion models, achieving performance comparable to autoregressive models and GANs. PriorGrad (gil Lee et al., 2022) enhances both speech quality and inference speed by employing a data-dependent prior distribution. Meanwhile, FreGrad (Nguyen et al., 2024) simplifies the model and reduces denoising time through the use of Discrete Wavelet Transform (DWT). Multi-Band Diffusion (MBD) (Roman et al., 2023) leverages a diffusion model to reconstruct waveforms from discrete EnCodec (Défossez et al., 2022) tokens. Despite these advancements, the current fastest generation speed of these methods is only about 10 to 20 times faster than real-time, which limits their application for real-time use, especially when combined with large-scale Transformer-based acoustic models (Wang et al., 2023; Du et al., 2024). In contrast, our method achieves speeds up to 160 times faster than real-time, significantly enhancing real-time applicability.

**Estimating Complex Spectrograms**  Waveform reconstruction from complex spectrograms can be effectively achieved using the ISTFT. Notably, Vocos (Siuzdak, 2023) and APNet2 (Du et al., 2023), utilizing GANs as their model framework, estimate magnitude and phase spectrograms from the input Mel-spectrograms, which can be transformed to complex spectrograms effortlessly. Both models operate at the frame level, enabling them to achieve significantly faster inference speeds compared to HiFi-GAN (Kong et al., 2020a), which uses multiple upsampling layers and operates at the level of waveform sample points. Morever, these models preserve the quality of the synthesized waveform, demonstrating their superiority in both speed and fidelity without a trade-off. In this paper, we directly estimate complex spectrograms using Rectified Flow and focus on frame-level operations, aiming to enhance both the efficiency and quality of our waveform synthesis process. Notably, the distributions of real and imaginary parts of the complex spectrograms appear more homogeneous in comparison to the distributions magnitude and phase.

**Multi-band Audio Waveform Reconstruction**  Both Multi-band MelGAN (Yang et al., 2021) and Multi-band Diffusion (Roman et al., 2023) employ multi-band strategies for different purposes. Multi-band MelGAN uses Pseudo-Quadrature Mirror Filters (PQMF) (Johnston, 1980) to divide frequency bands. This division results in each subband's waveform being a fraction of the original waveform's length, based on the number of subbands. By reshaping these subbands into feature dimensions and using a unified backbone, it operates on shorter signals, enhancing efficiency during training and inference. Multi-band Diffusion instead uses band-pass filters to separate frequency bands and models each subband distinctly, preventing cross-band error propagation. Our approach simplifies frequency band division by directly selecting appropriate dimensions from complex spectrograms. We improve efficiency by modeling all subbands together in parallel with a single unified model, increasing processing speed while reducing error accumulation across subbands.

## 3 METHOD

Our model utilizes a multi-band Rectified Flow with a ConvNeXtV2 (Woo et al., 2023) backbone, offering two alternative modeling choices with noisy samples in time or frequency domain while the neural network consistently operates at the STFT frame level. With only 10 steps of sampling, the model is capable of producing high-quality waveforms. In this section, we present the structure of the multi-band Rectified Flow model, describe the corresponding normalization techniques, three enhanced loss functions, and the strategy for selecting sampling time points.

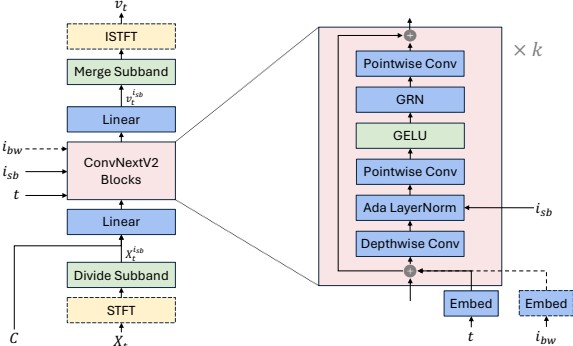

Figure 1: The overall structure for RFWave. $i_{sb}$ is the subband index, $C$ is the conditional input, which can be an Encodec token or Mel-spectrogram, and $i_{bw}$ is the EnCodec bandwidth index. Modules enclosed in a dashed box, as well as dashed arrows, are considered optional.

## 3.1 Multi-band Rectified Flow

**Model Structure**   The model structure is depicted in Figure 1. All frequency bands, each distinguished by a unique subband index, share the same model. The subbands of a given sample are grouped together into a single batch for processing, which facilitates simultaneous training or inference. This significantly reduces inference latency. Moreover, independently modeling the subbands reduces error accumulation. As discussed in (Roman et al., 2023), conditioning higher bands on lower ones can lead to an error accumulation, which means inaccuracies in the lower bands can adversely affect the higher bands during inference.

The model maps a noisy sample ($X_t$) to its velocity ($v_t$). For each subband, the subband's noisy sample ($X_t^{i_{sb}}$) is fed into the ConvNeXtV2 backbone to predict its velocity ($v_t^{i_{sb}}$) conditioned on time ($t$), the subband index ($i_{sb}$), the conditional input ($C$, the Mel-spectrogram or the EnCodec (Défossez et al., 2022) tokens), and an optional EnCodec bandwidth index ($i_{bw}$). The detailed structure of the ConvNeXtV2 backbone is shown in Figure 1. We employ Fourier features as described in (Kingma et al., 2021). The $X_t^{i_{sb}}$, $C$, and Fourier features are concatenated along the channel dimension and then passed through a linear layer, forming the input that is fed into a series of ConvNeXtV2 blocks. The sinusoidal $t$ embedding, along with the optional $i_{bw}$ embedding, are element-wise added to the input of each ConvNeXtV2 block. The $i_{bw}$ is utilized during the decoding of EnCodec tokens, enabling a single model to support EnCodec tokens with various bandwidths. Furthermore, the $i_{sb}$ is incorporated via an adaptive layer normalization module, which utilizes learnable embeddings as described in (Siuzdak, 2023; Xu et al., 2019). The other components are identical to those within the ConvNeXtV2 architecture, details can be found in (Woo et al., 2023).

Our methodology offers two modeling options. The first involves mapping Gaussian noise to the waveform directly in the time domain, wherein $X_0$, $X_1$, $X_t$ and $v_t$ all reside in the time domain. The second option maps Gaussian noise to the complex spectrogram, placing $X_0$, $X_1$, $X_t$ and $v_t$ in the frequency domain. Notably, $X_t^{i_{sb}}$ and $v_t^{i_{sb}}$ are consistently represented in the frequency domain, as detailed in the following paragraphs, ensuring that the neural network runs at the frame level. By processing frame-level features, our model achieves greater memory efficiency compared to diffusion vocoders like PriorGrad, which operate at the level of waveform sample points. While PriorGrad (gil Lee et al., 2022) can train on 6-second[2] audio clips at 44.1 kHz within 30 GB of GPU memory, our model is capable of handling 177-second clips with the same memory resources.

**Operating with $X_t$ in the Time Domain and Waveform Equalization**   Since our model is designed to function at the frame level, when $X_t$ and $v_t$ are in the time domain (specially, $X_1$ is the waveform and $X_0$ is noise of the identical shape, with $X_t$ and $v_t$ derived from (3)), the use of STFT and ISTFT, as illustrated in Figure 1, becomes necessary. The dimension of $X_t$ and $v_t$ adhere to $[1, T]^3$, where $T$ is the waveform length in sample points. After the STFT operation, we extract the subbands by equally dividing the full-band complex spectrogram, getting $X_t^{i_{sb}}$. Each subband $X_t^{i_{sb}}$

---

[2]The duration of an audio clip is calculated as the batch size multiplied by the number of segment frames and hop length, divided by the sampling rate.

[3]For simplicity, the batch dimension is not included in the discussion.

is then processed independently by the backbone as an individual sample to predict the corresponding $v_t^{i_{sb}}$. These predictions are subsequently merged back together before the ISTFT operation. The $X_t^{i_{sb}}$ processed by the backbone thus has dimensions of $[2d_s, F]$, where $d_s$ represents the number of frequency bins in a subband's complex spectrum and $F$ the number of frames. The real and imaginary parts of each subband are interleaved to form a $2d_s$-dimensional feature.

White Gaussian noise has uniform energy across frequency bands, but waveform energy profiles vary significantly. For example, speech energy decays exponentially with frequency, while music maintains a consistent distribution (Schnupp et al., 2011). These differences challenge diffusion model training, making energy equalization across bands beneficial (Roman et al., 2023). In the time-domain model, a bank of Pseudo-Quadrature Mirror Filters (PQMF) is employed to decompose the input waveform into subbands. Subsequently, these subbands are equalized and then recombined to form the equalized waveform. The performance of the PQMF bank exhibits a modest enhancement compared to the array of band-pass filters employed in (Roman et al., 2023). It's important to note that the PQMF is solely utilized for waveform equalization and holds no association with the division of the complex spectrogram into subbands. For waveform equalization, mean-variance normalization is employed, utilizing the exponential moving average of mean and variance of each waveform subband computed during training. This approach ensures that the transformation can be effectively inverted using the same statistics.

**Alternative Approach: Operating with $X_t$ in the Frequency Domain and STFT Normalization**
When $X_t$ and $v_t$ reside in the frequency domain (i.e., $X_1$ is the waveform's complex spectrogram and $X_0$ is noise of the identical shape), STFT and ISTFT, as shown in Figure 1, are unnecessary. The dimensions of $X_t$ and $v_t$ are $[2d, F]$, where $d$ is the number of frequency bins in the complex spectrogram. By equally partitioning the full-band complex spectrogram, $X_t^{i_{sb}}$ is extracted, resulting in a shape of $[2d_s, F]$, which is then processed by the ConvNeXtV2 backbone. In the frequency-domain model, the waveform is transformed to a complex spectrogram without equalization. The preprocessing involves the dimension-wise mean-variance normalization of the complex spectrogram.

During inference, operating with $X_t$ and $v_t$ in the time domain requires both STFT and ISTFT at each sampling step. In contrast, when $X_t$ and $v_t$ are in the frequency domain, only a single ISTFT is needed after the entire sampling process. Despite this computational overhead, our experiments show that the time-domain configuration achieves slightly better performance, particularly in preserving high-frequency details. Sampling algorithms for the two distinct approaches, one in the time domain and the other in the frequency domain, are provided in Appendix Section A.9.1.

### 3.2 LOSS FUNCTIONS

**Energy-balanced Loss**   In preliminary experiments, we noticed low-volume noise in expected silent regions. We attribute this to the property of Mean Square Error (MSE) used in (2). The MSE measures the absolute distortion between the predicted values and the ground truth. In silent regions, small absolute errors contribute minimally to the overall MSE loss, so the model does not prioritize eliminating them during training. This results in the model's inability to effectively suppress minor deviations in silent areas, potentially leading to perceptible noise. In contrast, larger errors in high-amplitude regions have a significant impact on the MSE loss, causing the model to focus more on reducing errors in these areas during training.

We propose energy-balanced loss to mitigate this problem. Our energy-balanced loss is designed to weight errors differently depending on the region's volume (or energy) accross the time-axis. Specifically, for each frequency subband, we compute the standard deviation along the feature dimension of the ground truth velocity to construct a weighting coefficient of size $[1, F]$. This vector is reflective of the frame-level energy of the respective subband, as depicted in Figure A.1. Subsequently, both the ground truth and predicted velocity are divided by this vector before proceeding to the subsequent steps. For the frequency domain model, the training objective defined in (2) is adjusted as follows:

$$\min_v \mathbb{E}_{X_0 \sim \pi_0, (X_1, \mathcal{C}) \sim D} \left[ \int_0^1 || (X_1 - X_0)/\sigma - v(X_t, t \mid \mathcal{C})/\sigma ||^2 \, \mathrm{d}t \right], \tag{5}$$

$$\text{with } \sigma = \sqrt{\mathrm{Var}_1(X_1 - X_0)} \quad \text{and} \quad X_t = tX_1 + (1-t)X_0,$$

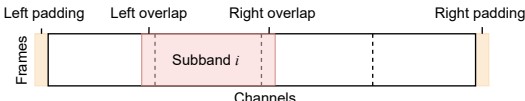

Figure 2: An illustration of dividing complex spectrograms into subbands. The area highlighted in pink represents a subband, while the section enclosed by the two dashed vertical lines indicates the main section.

where $D$ represents the dataset with paired $X_1$ and $\mathcal{C}$, and $\text{Var}_1$ calculates the variance along the feature dimension. For the time domain model, this energy balancing operation precedes the ISTFT process. This approach helps to minimize the relative error in low-volume regions. Our experimental results demonstrate that this method enhances overall performance, benefiting not just the silent parts.

**Overlap Loss**  Within the multi-band structure, each subband is predicted independently, potentially resulting in inconsistencies among them. To mitigate these inconsistencies, we introduce an overlap loss. This involves maintaining overlaps between the subbands when dividing the full-band complex spectrograms. A detailed illustration of this scheme is provided in Figure 2 and described in detail in Appendix Section A.3. In this paper, we employ an 8-dimensional overlap. During the training phase, the MSEs of the overlapped predictions is minimized. In the inference phase, the overlaps are removed, and all subbands are merged to recreate the full-band complex spectrograms.

As each subband is predicted, the model internally maintains consistency between its overlap section and the main section. The overlap serves as an anchor to maintain consistency among the subbands. Modeling higher bands based on lower bands rather than predicting all subbands in parallel can increase consistency between subbands. However, at the inference stage, if the lower bands are incorrectly predicted, the higher bands conditioned on them would also be inaccurate (Roman et al., 2023). RFWave addresses this by using overlap loss to maintain consistency between subbands during training. At the inference stage, subbands are predicted independently, so any error occurring in one subband does not negatively affect the others.

**STFT Loss**  The magnitude spectrogram derived loss (Arik et al., 2018) is extensively utilized in GAN-based vocoders, such as HiFi-GAN (Kong et al., 2020a) and Vocos (Siuzdak, 2023), both of which leverage a Mel-spectrogram loss. Nevertheless, its use in diffusion-based vocoders is relatively rare. This might stem from its lack of direct compatibility with the formalization of a noise prediction diffusion model. Here we adopt STFT loss for RFWave. According to (5), the model's output serves as an approximation for velocity:

$$v(X_t, t \mid \mathcal{C}) \approx \frac{\mathrm{d}}{\mathrm{d}t} X_t = X_1 - X_0. \tag{6}$$

Hence, at time $t$, an approximation for $X_1$ is:

$$\widetilde{X_1} \approx X_0 + v(X_t, t \mid \mathcal{C}). \tag{7}$$

The STFT loss can be applied on the approximation $\widetilde{X_1}$, incorporating the spectral convergence loss and log-scale STFT-magnitude loss as proposed in (Arik et al., 2018), which are detailed in Appendix Section A.4. According to our experimental findings, the STFT loss effectively reduces artifacts in the presence of background noise.

### 3.3 SELECTING TIME POINTS FOR EULER METHOD

Liu et al. (2023) employ equal step intervals for the Euler method, as illustrated in (4). Here, we propose selecting time points for sampling based on the straightness of transport trajectories. The time points are chosen such that the increase in straightness is equal across each step. The straightness of a learned velocity field $v$ is defined as $S(v) = \int_0^1 \mathbb{E} \parallel (X_1 - X_0) - v(X_t, t \mid \mathcal{C}) \parallel^2 \mathrm{d}t$, which describes the deviation of the velocity along the trajectory. A smaller $S(v)$ means straighter trajectories. This approach ensures that the difficulty of each Euler step remains consistent, requiring the model to take more steps in more challenging regions. Using the same number of sampling steps, this method outperforms the equal interval approach. The implementation details are provided in Appendix Section A.5, while the algorithm is presented in Appendix Section A.9.2. We refer to this approach as **equal straightness**.

## 4 EXPERIMENTS AND ANALYSIS

### 4.1 EXPERIMENTAL SETUP

**Overview**   We evaluate RFWave using both Mel-spectrograms and discrete EnCodec tokens as inputs. For Mel-spectrogram inputs, we first benchmark RFWave against existing diffusion vocoders to demonstrate its superiority. We then compare it with widely-used GAN models to highlight its practical applicability and advantages. For discrete EnCodec tokens, we evaluate RFWave's efficiency in reconstructing high-quality audio from compressed representations across diverse domains. Finally, we conduct ablation studies and further analysis to examine the effects of the individual components of RFWave.

**Data**   For Mel-spectrogram inputs, we conduct two evaluations. When benchmarking against diffusion vocoders, we train separate models on LibriTTS (Zen et al., 2019) (speech), MTG-Jamendo (Bogdanov et al., 2019) (music), and Opencpop (Wang et al., 2022) (vocal) datasets and test each model on its respective dataset to ensure comprehensive comparison across various audio categories. When comparing RFWave to widely used GAN-based models, we train a model on LibriTTS and evaluate its in-domain performance on the LibriTTS test set. Additionally, we assess the out-of-domain generalization ability of this LibriTTS-trained model by testing it on the MUSDB18 (Rafii et al., 2017) test subset.

For discrete EnCodec token inputs, we follow convention by training a universal model on a large-scale dataset. This dataset combines Common Voice 7.0 (Ardila et al., 2019) and clean data from DNS Challenge 4 (Dubey et al., 2022) for speech, MTG-Jamendo (Bogdanov et al., 2019) for music, and FSD50K (Fonseca et al., 2021) and AudioSet (Gemmeke et al., 2017) for environmental sounds. Recognizing the lack of a comprehensive test set for universal audio codec models, we constructed a unified evaluation dataset comprising 900 test audio samples from 15 external datasets, covering speech, vocals, and sound effects. Detailed information about this test set is provided in Table A.4.

**Baseline and Evaluation Metrics**   We conducted a comprehensive benchmark comparison of RFWave against a series of state-of-the-art models. For Mel-spectrogram inputs, we utilized PriorGrad (gil Lee et al., 2022) and FreGrad (Nguyen et al., 2024) as baselines for diffusion-based models. Vocos (Siuzdak, 2023) and BigVGAN (gil Lee et al., 2023) served as baselines for GAN-based methods, reflecting their widespread use in real-world applications. For discrete EnCodec token inputs, we compared with EnCodec (Défossez et al., 2022) and MBD (Roman et al., 2023).

We trained PriorGrad and FreGrad on LibriTTS, Opencpop, and MTG-Jamendo using their open-source details. For Vocos, BigVGAN, EnCodec, and MBD, we used public pre-trained models. We followed the authors' recommended sampling steps: 6 for PriorGrad, 50 for FreGrad, and 20 for MBD. Model sources are in Table A.2.

For objective evaluation, we use ViSQOL (Chinen et al., 2020) to assess perceptual quality, employing speech-mode for 22.05/24 kHz waveforms and audio-mode for 44.1 kHz waveforms. Additional metrics include the Perceptual Evaluation of Speech Quality (PESQ) (Rix et al., 2001), the F1 score for voiced/unvoiced classification (V/UV F1), and the periodicity error (Periodicity) (Morrison et al., 2022). For subjective evaluation, we conduct crowd-sourced assessments, employing a 5-point Mean Opinion Score (MOS) to determine the audio naturalness, ranging from 1 ('poor/unnatural') to 5 ('excellent/natural'), more details provided in the Appendix Section A.10.

**Implementation**   We use the time-domain model with three enhanced loss functions as the default. The RFWave backbone contains 8 ConvNeXtV2 blocks, and the complex spectrogram is divided into 8 equally spanned subbands. For evaluation, we use 10 sampling steps. Table A.3 lists the parameters used for extracting Mel-spectrograms and complex spectrograms from datasets with different sample rates. Further implementation details and computational resource requirements are provided in the Appendix Sections A.1 and A.2, respectively.

## 4.2 EXPERIMENTAL RESULTS ON MEL-SPECTROGRAMS INPUT

**Comparison with Diffusion-based Method**   In this portion, we compare RFWave[4] with PriorGrad and FreGrad by training separate models for each method on the LibriTTS, MTG-Jamendo, and Opencpop datasets and evaluating each model on its respective test set. Table 1 presents overall results, while Table A.5 provides detailed metrics for each audio category. RFWave consistently outperforms PriorGrad and FreGrad in both objective and subjective metrics. The superior MOS score achieved by RFWave is likely due to its ability to produce clearer and more consistent harmonics, particularly in high-frequency ranges, which contributes to better overall audio quality. A collection of spectrograms and their corresponding analyses can be found in Appendix Section A.6.

**Comparison with GAN-based Method**   When comparing RFWave to widely used state-of-the-art GAN-based models (BigVGAN, Vocos), we used models trained on LibriTTS. Besides evaluating on the LibriTTS testset, to assess the models' robustness and extrapolation capabilities, following the methodology in (gil Lee et al., 2023), we also evaluated these models' performance on the MUSDB18 dataset. It can be observed in Table 2 that on the in-domain test set (LibriTTS), RFWave is generally on par with state-of-the-art GAN-based models. However, on the out-of-domain test set (MUSDB18), according to the results in Table 3, RFWave shows significant advantages over BigVGAN and Vocos, even though BigVGAN employed Snake activation (Ziyin et al., 2020) to enhance its out-of-domain data generation capabilities. This demonstrates that RFWave, as a diffusion-type model, offers clear advantages in generalization and robustness compared to GAN-based models. For detail, we observe that GAN-based methods tend to generate horizontal lines in the high-frequency regions of the spectrograms, as exemplified in Figure A.7. In contrast, RFWave consistently produces clear high-frequency harmonics, even when applied to out-of-domain data.

Table 1: Average Mean Opinion Score (MOS) and objective evaluation metrics for RFWave, PriorGrad and FreGrad across various test sets. MOS is provided with 95% confidence interval.

| Model | MOS ↑ | PESQ ↑ | ViSQOL ↑ | V/UV F1 ↑ | Periodicity ↓ |
|---|---|---|---|---|---|
| RFWave | **3.95**±0.09 | **4.202** | **4.456** | **0.979** | **0.070** |
| PriorGrad | 3.75±0.09 | 3.612 | 4.347 | 0.974 | 0.082 |
| FreGrad | 2.99±0.14 | 3.640 | 4.179 | 0.973 | 0.087 |
| Ground truth | 4.00±0.09 | - | - | - | - |

Table 2: MOS and objective evaluation metrics for RFWave, BigVGAN and Vocos on LibriTTS.

| Model | MOS ↑ | PESQ ↑ | ViSQOL ↑ | V/UV F1 ↑ | Periodicity ↓ |
|---|---|---|---|---|---|
| RFWave | **3.82**±0.12 | **4.304** | 4.579 | 0.967 | 0.091 |
| BigVGAN | 3.78±0.11 | 4.240 | **4.712** | **0.978** | **0.067** |
| Vocos | 3.74±0.10 | 3.660 | 4.696 | 0.958 | 0.104 |
| Ground truth | 3.91±0.10 | - | - | - | - |

Table 3: MOS for RFWave, BigVGAN and Vocos on MUSDB18.

| Model | Vocals | Drums | Bass | Others | Mixture | Average |
|---|---|---|---|---|---|---|
| RFWave | **3.46**±0.14 | 3.65±0.10 | **3.62**±0.10 | **3.54**±0.11 | **4.04**±0.11 | **3.67**±0.05 |
| BigVGAN | 3.42±0.13 | **3.68**±0.09 | 3.50±0.12 | 3.33±0.13 | 3.58±0.10 | 3.51±0.05 |
| Vocos | 3.04±0.15 | 3.54±0.11 | 2.96±0.12 | 2.88±0.15 | 3.08±0.14 | 3.10±0.06 |
| Ground truth | 3.62±0.14 | 3.71±0.14 | 3.75±0.12 | 3.79±0.12 | 4.17±0.10 | 3.80±0.05 |

## 4.3 EXPERIMENTAL RESULTS ON DISCRETE ENCODEC TOKENS INPUT

We evaluated RFWave against EnCodec and MBD across various bandwidths for discrete EnCodec tokens inputs. For RFWave, we used Classifier-Free Guidance (CFG) (Ho & Salimans, 2021) with

---

[4]This evaluation used an earlier RFWave version without equal straightness from Section 3.3. Despite this, the results convincingly demonstrate RFWave's superiority over other diffusion vocoders.

a 2.0 coefficient. CFG showed little improvement for Mel-spectrogram inputs, but effective for EnCodec tokens, likely due to CFG tending to be more impactful when the input is more compressed.

Table 4 shows average MOS and objective metrics for EnCodec tokens across bandwidths and auditory contexts, while Table A.6 provides detailed results for each category. RFWave achieves optimal scores in all metrics except ViSQOL, where EnCodec's GAN-based decoder excels. ViSQOL shows a subtle bias toward GAN-based models, likely due to distinct waveform footprints. While using a larger bandwidth generally improves performance, increasing bandwidth from 6.0 to 12.0 kbps slightly improves MOS for RFWave.

Table 4: Average MOS and objective metrics for RFWave, EnCodec and MBD across various test sets.

| Bandwidth | Model | MOS ↑ | PESQ ↑ | ViSQOL ↑ | V/UV F1 ↑ | Periodicity↓ |
|---|---|---|---|---|---|---|
| 1.5 kbps | RFWave$_{(CFG2)}$ | **3.17**±0.22 | **1.797** | 3.108 | **0.914** | **0.193** |
|  | EnCodec | 2.23±0.23 | 1.708 | **3.518** | 0.906 | 0.199 |
|  | MBD | 3.01±0.19 | 1.699 | 2.982 | 0.901 | 0.212 |
| 3.0 kbps | RFWave$_{(CFG2)}$ | **3.52**±0.25 | **2.444** | 3.570 | **0.939** | **0.145** |
|  | EnCodec | 2.79±0.25 | 1.934 | **3.793** | 0.930 | 0.166 |
|  | MBD | 3.06±0.23 | 2.310 | 3.402 | 0.922 | 0.171 |
| 6.0 kbps | RFWave$_{(CFG2)}$ | **3.69**±0.16 | **2.936** | 3.892 | **0.954** | **0.117** |
|  | EnCodec | 3.10±0.15 | 2.432 | **4.091** | 0.951 | 0.126 |
|  | MBD | 3.43±0.15 | 2.488 | 3.582 | 0.929 | 0.168 |
| 12.0 kbps | RFWave$_{(CFG2)}$ | **3.73**±0.16 | **3.270** | 4.124 | **0.965** | **0.099** |
|  | EnCodec | 3.55±0.15 | 2.892 | **4.291** | 0.963 | 0.105 |
|  | MBD | - | - | - | - | - |
| Ground truth |  | 4.07±0.14 | - | - | - | - |

## 4.4 ANALYSIS

**Ablations** We evaluate the model in frequency and time domains, then analyze the impact of incrementally adding the three loss functions and equal straightness to the better-performing model. Ablation studies on LJSpeech (Ito & Johnson, 2017) (250 test sentences) are summarized in Table 5. Despite the metrics not consistently correlating with human evaluations, they accurately captured the quality improvements brought about by design modifications (Roman et al., 2023). The model operating in the time domain outperforms its frequency domain counterpart, as the ISTFT operation in the former (Figure 1) introduces periodic signals. The inverse Discrete Fourier Transform (IDFT) matrix comprises sinusoidal functions of frequencies $[0, 1/N, ..., (N-1)/N]$, where $N$ is the number of FFT points (Oppenheim & Schafer, 1975). This mechanism is similar to the Snake activation (Ziyin et al., 2020) in BigVGAN (gil Lee et al., 2023), which also introduces periodic signals into the generator. Energy-balanced loss and equal straightness improves the overall performance. STFT loss improves PESQ but degrades ViSQOL and periodicity due to its emphasis on magnitude over phase. This trade-off is essential for effectively eliminating artifacts, especially in the presence of background noise. While overlap loss slightly lowers PESQ, it ensures smooth subband transitions. The effects are visualized in Figures A.8 and A.9.

Table 5: Objective metrics assess the model's performance in the frequency domain and the time domain, where three loss functions and equal straightness are applied incrementally.

| Setting | PESQ ↑ | ViSQOL ↑ | V/UV F1 ↑ | Periodicity ↓ |
|---|---|---|---|---|
| frequency | 3.872 | 4.430 | 0.948 | 0.144 |
| time | 4.127 | 4.551 | 0.957 | 0.124 |
| + energy-balanced loss | 4.181 | 4.598 | 0.965 | 0.110 |
| + overlap loss | 4.158 | 4.599 | 0.959 | 0.122 |
| + STFT loss | 4.211 | 4.578 | 0.961 | 0.119 |
| + equal straightness | 4.275 | 4.654 | 0.966 | 0.100 |

**Futher Analysis** Using the model with an 8-layer ConvNeXtV2 backbone, time domain, and three loss functions as the baseline, we first experimented with the DDPM (Ho et al., 2020) approach.

Utilizing the noise schedule from DiffWave and PriorGrad, the performance with 50 sampling steps fell short of the baseline, as demonstrated in Table 6. This highlights that **Rectified Flow is critical for our model to efficiently generate high-quality audio samples**. Next, we experimented with a ResNet (He et al., 2016) backbone with a similar number of parameters. This configuration slightly reduced efficiency and performance in objective metrics (Table 6), indicating that the **ConvNeXtV2 backbone enhances both efficiency and audio quality**. Finally, we experimented with models of different sizes, testing ConvNextV2 backbones at half and double sizes, with configurations in A.1. **Increasing backbone size improves performance but reduces efficiency** (Table 6).

Table 6: Objective metrics for further analysis. xRT stands for the speed at which the model can generate speech in comparison to real-time.

| Setting | GPU xRT ↑ | PESQ ↑ | ViSQOL ↑ | V/UV F1 ↑ | Periodicity ↓ |
|---|---|---|---|---|---|
| baseline | 162.59 | 4.211 | 4.578 | 0.961 | 0.119 |
| DDPM, 50 steps | 34.40 | 2.790 | 4.499 | 0.939 | 0.174 |
| ResNet | 150.95 | 3.865 | 4.509 | 0.955 | 0.128 |
| half size | 228.70 | 4.128 | 4.553 | 0.959 | 0.120 |
| double size | 111.62 | 4.285 | 4.636 | 0.968 | 0.100 |

## 4.5 INFERENCE SPEED

We perform inference speed benchmark tests using an NVIDIA GeForce RTX 4090 GPU. The implementation was done in PyTorch (Paszke et al., 2019), and no specific hardware optimizations were applied. The inference was carried out with a batch size of 1 sample, utilizing the LJSpeech test set, resampled to model's sampling rate. Table 7 displays the model size, synthesis speed and GPU memory consumption of the models. RFWave is more than twice as fast as BigVGAN and consumes less GPU memory, thereby eliminating latency as a barrier for practical applications. This speed advantage becomes even more pronounced when synthesizing high-resolution audio (44.1/48 kHz). Frame-level models like RFWave significantly outperform sample-point-based models in high-resolution audio synthesis in terms of speed. RFWave can easily adjust the complex spectrogram's window length and hop length for 44.1 kHz sampling, maintaining low computational complexity, whereas BigVGAN requires additional upsampling blocks. Vocos serves as a strong baseline, given its efficiency in requiring only a single forward pass and operating at the frame level.

Table 7: Model footprint and synthesis speed. xRT stands for the speed at which the model can generate speech in comparison to real-time.

| Model | Parameters (M) | GPU xRT ↑ | GPU Memory (MB) | Sampling steps |
|---|---|---|---|---|
| RFWave | 18.1 | 162.59 | 780 | 10 |
| BigVGAN | 107.7 | 72.68 | 1436 | 1 |
| Vocos(ISTFT) | 13.5 | **2078.20** | **590** | 1 |
| PriorGrad | 2.6 | 16.67 | 4976 | 6 |
| FreGrad | **1.7** | 7.50 | 2720 | 50 |
| MBD | 411.0 | 4.82 | 5480 | 20 |
| RFWave(44.1KHz) | 20.5 | 152.58 | 902 | 10 |
| BigVGAN(44.1KHz) | 116.5 | 39.03 | 1740 | 1 |

## 5 CONCLUSION

In this study, we propose RFWave, a multi-band Rectified Flow approach for audio waveform reconstruction. The model has been carefully designed to overcome the latency issues associated with diffusion models. RFWave stands out for its ability to generate complex spectrograms by operating at the frame level, processing all subbands concurrently. This concurrent processing significantly enhances the efficiency of the waveform reconstruction process. The empirical evaluations conducted in this research have demonstrated that RFWave achieves exceptional reconstruction quality. Moreover, it has shown superior computational efficiency by generating audio at a speed that is 160 times faster than real-time, comparable to GAN-based methods, making it practical for real-world applications.

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

# A    APPENDIX

## A.1    IMPLEMENTATION DETAILS

The RFWave backbone contains 8 ConvNeXtV2 blocks. Within each ConvNeXtV2 block, the depth-wise convolutional layer featuring a large kernel utilizes a kernel size of 7 and has a channel dimension of 512. The first and last 1x1 point-wise convolutional layers in the sequence possess channel dimensions of 512 and 1536, respectively. In Subsection 4.4, the half-size variant contains 8 ConvNeXtV2 blocks with point-wise convolutional layers of 384 and 1152, while the double-size variant features 16 ConvNeXtV2 blocks with point-wise convolutional layers of 512 and 1536.

During the extraction of complex coefficients for the model, we use the orthonormal Fast Fourier Transform (FFT) and its inverse (IFFT), with the normalization convention of dividing by $1/\sqrt{N}$ for both operations, here $N$ is the FFT size. This approach ensures the spectrogram extracted is within a more reasonable range for modeling.

In terms of loss functions, we assign a weight of 1 to either the Rectified Flow loss or its energy-balanced variant. If employed, the overlap loss and STFT loss are assigned a weight of 0.01. These weights are determined by aligning the L2-norm of the gradients from the overlap loss and STFT loss with approximately 1/10 of that from the Rectified Flow loss.

Training details are also worth mentioning. Audio samples are randomly cropped to lengths of 32512 and 65024 for 22.05/24 kHz and 44.1 kHz waveforms, respectively. This is equivalent to a crop window of 128 frames for both sampling rates. We use a batch size of 64. The model optimization is performed using the AdamW optimizer with a starting learning rate of 2e-4 and beta parameters of (0.9, 0.999). A cosine annealing schedule is applied to reduce the learning rate to a minimum of 2e-6.

## A.2    COMPUTATIONAL RESOURCE REQUIRED FOR THE EXPERIMENTS

The bulk of the experiments were carried out on personal computers and GPU servers, which were sourced from cloud service providers. Primarily, we utilized Nvidia-4090-24G and Nvidia-A100-80G GPUs for these tasks.

The specific hardware used and the corresponding time taken for training RFWave on various datasets are as follows:

Table A.1: GPU configurations and training duration for various datasets

| Dataset | GPU configuration | Training duration |
|---------|-------------------|-------------------|
| LJspeech | 1×4090 | 2 days |
| LibriTTS | 2×A100 | 5 days |
| Opencpop | 1×4090 | 1 day |
| MTG-Jamendo | 2×A100 | 7 days |
| EnCodec mixed dataset | 4×A100 | 10 days |

## A.3    DEVIDING INTO SUBBANDS

The complex spectrogram is circularly padded in the feature dimension with a size of $(d_{ol}, d_{ol} - 1)$, where $d_{ol}$ signifies the overlap size. The dimension of each subband's main section, denoted as $d_m$, is calculated by dividing $d - 1$ by the total number of subbands. Here, $d$ represents the dimension of the complex spectrograms. The last subband is an exception, having an extra feature dimension but one less padding dimension. Subbands are extracted by applying a sliding window along the feature dimension, where the window has a size of $d_m + 2d_{ol}$ and shifts by $d_m$[5].

---

[5]The feature dimension of each subband, represented as $d_s$, equals $(d_m + 2d_{ol})$, accounting for the interleave of real and imaginary parts.

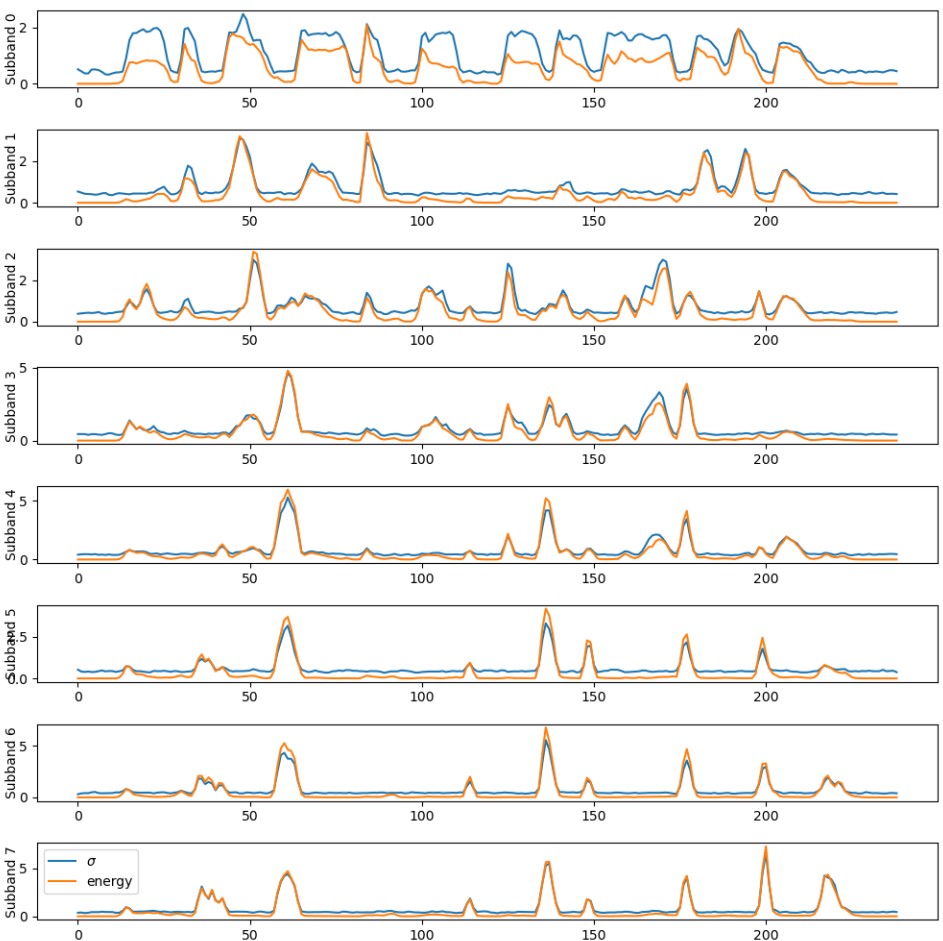

Figure A.1: The energy and weighting coefficients, represented by $\sigma$, display a consistent variation throughout the frames.

## A.4 STFT LOSS

**Spectral convergence (SC) loss**

$$\||X_1| - |\widetilde{X}_1|\|_F / \||X_1|\|_F, \tag{8}$$

where $X_1$ is the ground truth and $\widetilde{X}_1$ is the estimated complex spectrogram using (7). The Frobenius norm, $\|\cdot\|_F$, applied over time and frequency, emphasizes large spectral components in SC loss.

**Log-scale STFT-magnitude loss**

$$\|\log(|X_1| + \epsilon) - \log(|\widetilde{X}_1| + \epsilon)\|_1, \tag{9}$$

where the $L^1$ norm, $\|\cdot\|_1$, along with a small constant $\epsilon$, is used to accurately capture small-amplitude components in the log-scale STFT-magnitude loss.

## A.5 SELECTING TIME POINTS FOR EULER METHOD

Straightness is calculated from a single batch with a size of 96. The Euler method uses 100 equal interval steps to estimate the integral in the straightness definition. Time points are selected such that the increase in straightness remains consistent across each interval. An example with 10 Euler steps is provided in Figure A.2. The time points are calculated only once per model, resulting in negligible computational load.

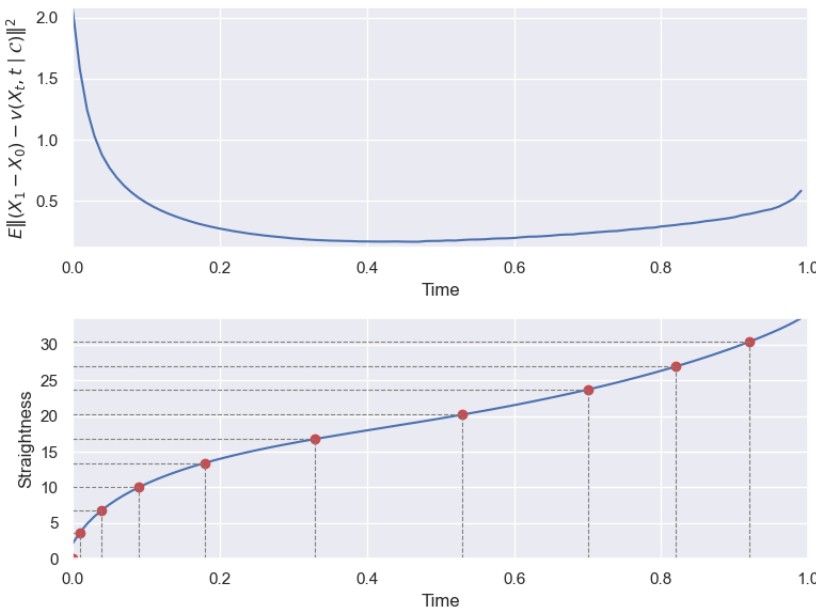

Figure A.2: Deviation (top) and straightness (bottom) over time: red dots mark Euler method time points, with constant increase in straightness across intervals.

A.6    SPECTROGRAM EXAMPLES FOR COMPARISON WITH DIFFUSION-BASED METHOD

For all the spectrogram figures, we used Adobe Audition for spectrogram visualization, with the dynamic range being [-132, 0] dB.

Figure A.3 showcases spectrogram examples generated by various models using the Opencpop dataset. When compared to the ground truth spectrogram, RFWave is seen to produce clean and stable harmonics, whereas the harmonics generated by other models exhibit minor discontinuities.

Figure A.4 exhibits spectrogram examples generated by different models utilizing the LibriTTS dataset. RFWave generate clear high-frequency harmonics, while PriorGrad and FreGrad result in blurred high-frequency harmonics.

Figure A.5 displays spectrogram examples generated by diverse models from the Jamendo dataset. Both RFWave and PriorGrad generate commendable spectrograms, with RFWave edging out slightly in the high-frequency range.

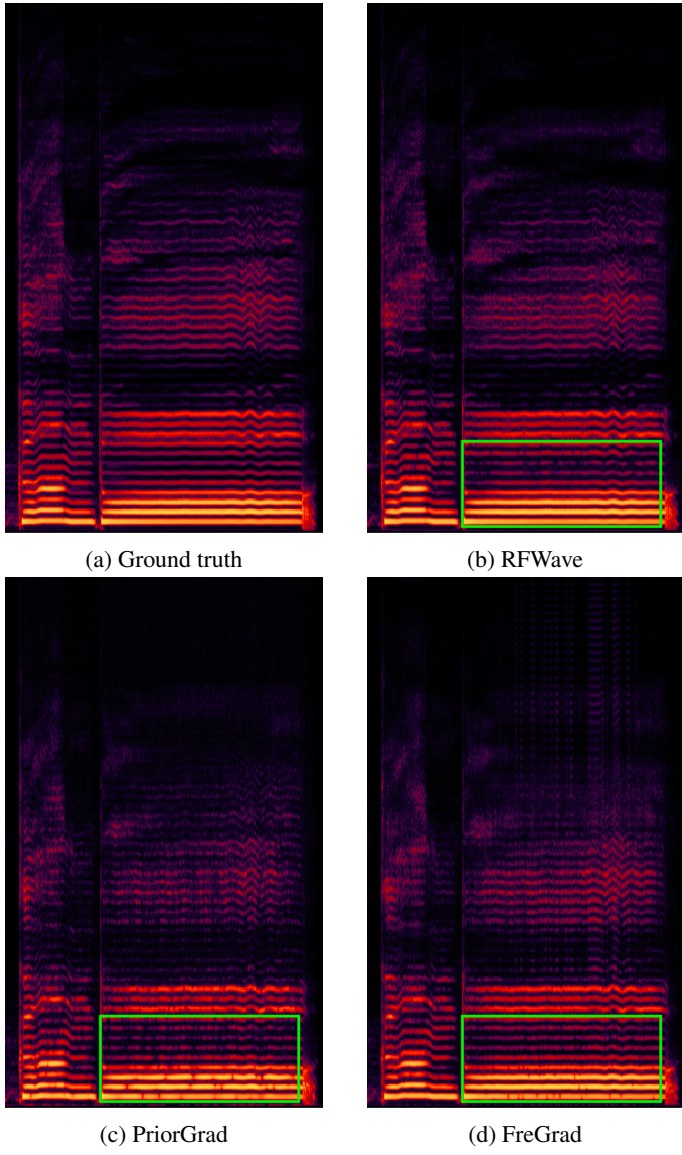

(a) Ground truth                    (b) RFWave

(c) PriorGrad                        (d) FreGrad

Figure A.3: Examples of spectrograms from Opencpop. Within the region marked out by the green box, the harmonics generated through PriorGrad and FreGrad show minor discontinuities.

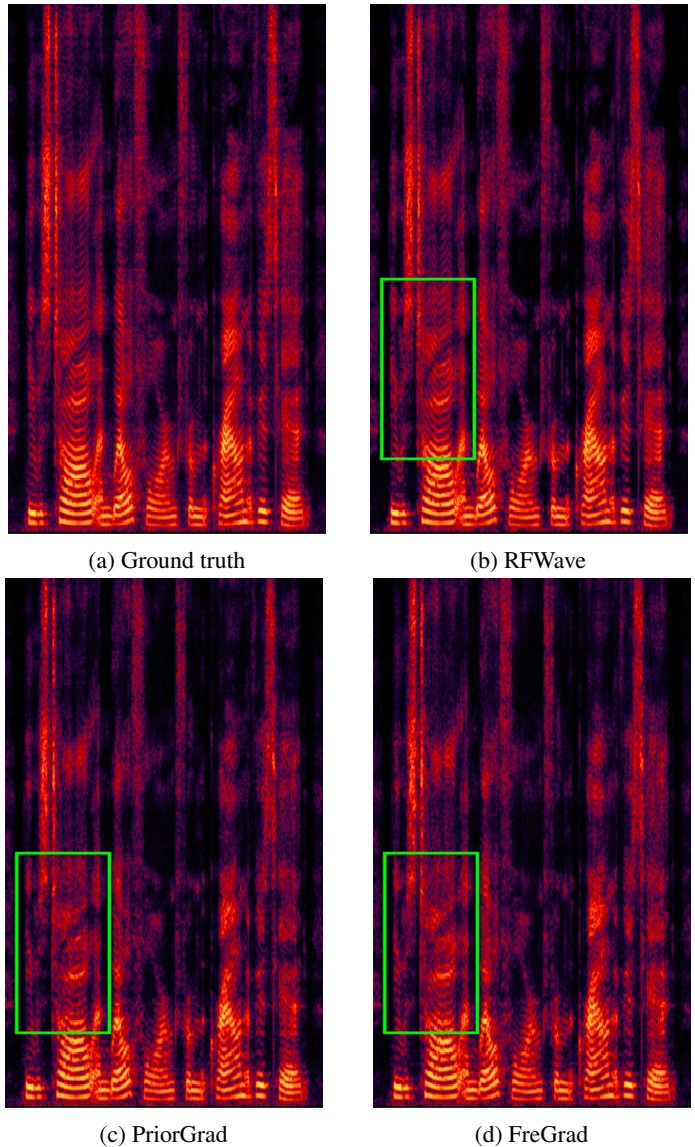

(a) Ground truth        (b) RFWave

(c) PriorGrad        (d) FreGrad

Figure A.4: Examples of spectrograms from LibriTTS. Within the green-boxed region, RFWave yields better harmonics than PriorGrad and FreGrad.

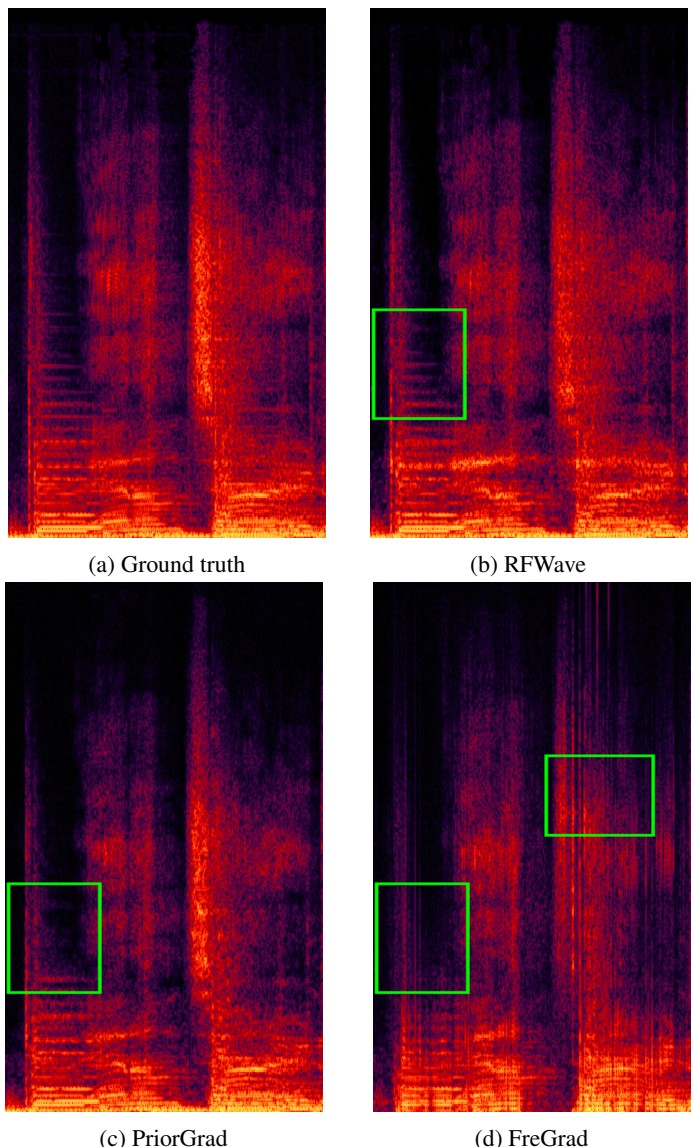

Figure A.5: Examples of spectrograms from MTG-Jamendo. Within the green-boxed region, RFWave generates more favorable high-frequency components compared to PriorGrad and FreGrad. Additionally, FreGrad gives rise to vertical line artifacts as highlighted by the top-right box.

## A.7 Spectrogram Examples for Comparison with GAN-based Method

Figure A.6 illustrates that RFWave, BigVGAN and Vocos are capable of generating high-quality spectrograms when applied to the LibriTTS dataset. This demonstrates the effectiveness of each approach in handling in-distribution data. The spectrograms exhibit well-defined harmonics and minimal artifacts, highlighting the robustness of the models within their familiar domain.

Figure A.7 presents spectrogram examples generated on the MUSDB18 dataset, with the models having been trained on the LibriTTS dataset. A notable observation is that both BigVGAN and Vocos exhibit a tendency to produce horizontal lines in the high-frequency regions of the spectrograms. These artifacts are perceptually problematic, as they often result in a metallic sound quality that detracts from the naturalness of the audio. In stark contrast, RFWave consistently generates clear and well-defined high-frequency harmonics. This capability underscores RFWave's superior ability to maintain spectral fidelity, even when applied to out-of-distribution data, thereby enhancing the overall audio quality and realism.

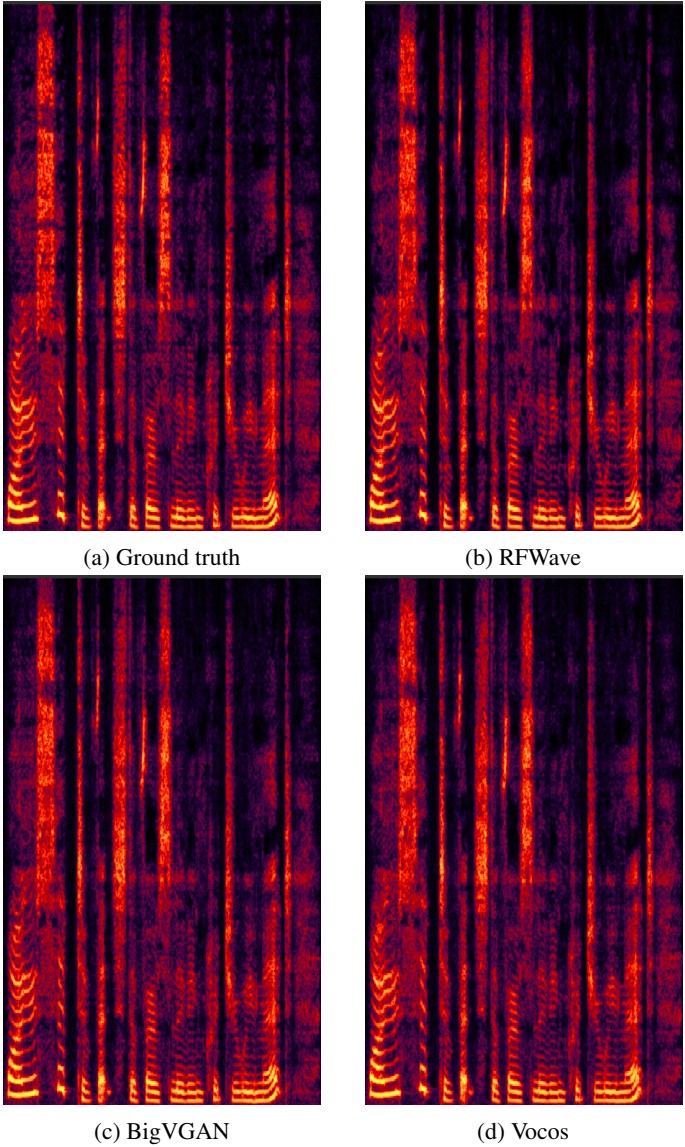

|  |  |
|:---:|:---:|
| (a) Ground truth | (b) RFWave |
| (c) BigVGAN | (d) Vocos |

Figure A.6: Examples of spectrograms from LibriTTS. All the methods produce high-quality spectrograms.

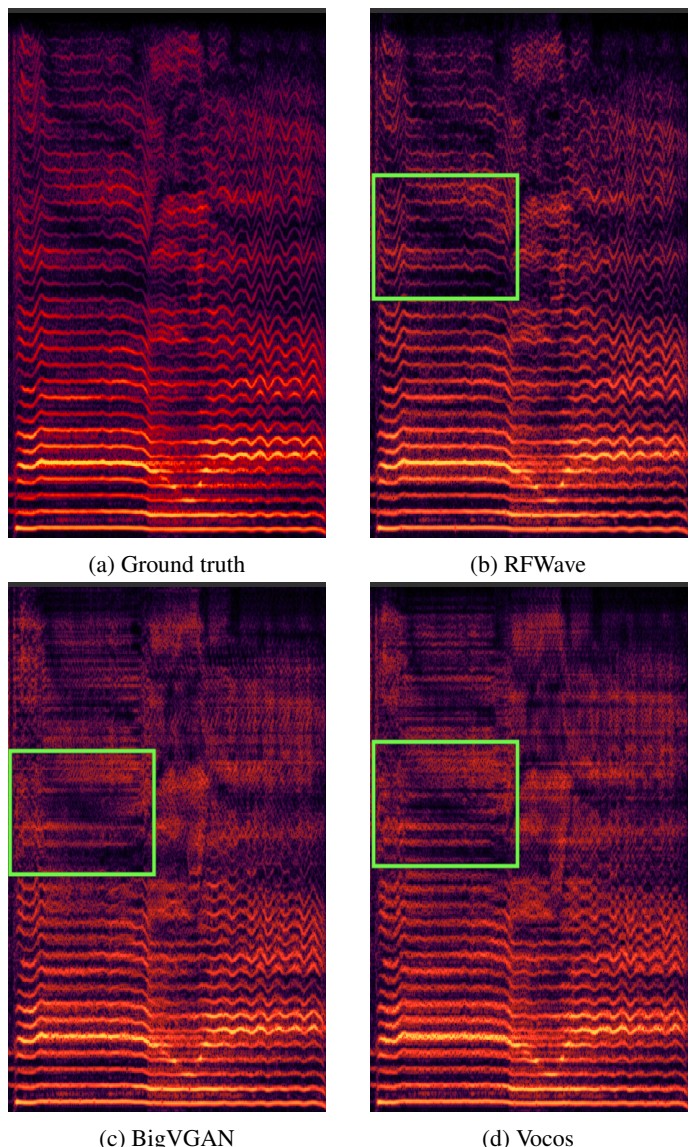

(a) Ground truth

(b) RFWave

(c) BigVGAN

(d) Vocos

Figure A.7: Examples of spectrograms from MUSDB18. Within the green-boxed region, both BigVGAN and Vocos have a propensity to generate horizontal lines within the high-frequency areas of the spectrograms. These artifacts present perceptual issues since they frequently lead to a metallic sound quality that diminishes the naturalness of the audio. In sharp contrast, RFWave produces well defined high-frequency harmonics, even when applied to out-of-distribution data.

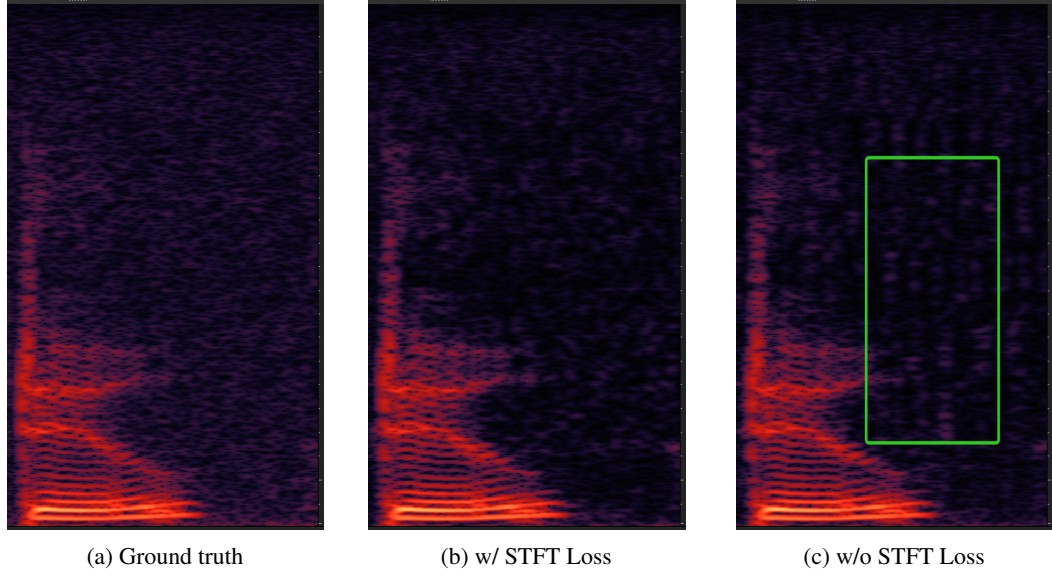

(a) Ground truth   (b) w/ STFT Loss   (c) w/o STFT Loss

Figure A.8: The effect of STFT loss. There are some vertical patterns in the spectrogram of waveforms generated by a model without STFT loss, as highlighted by the green rectangular.

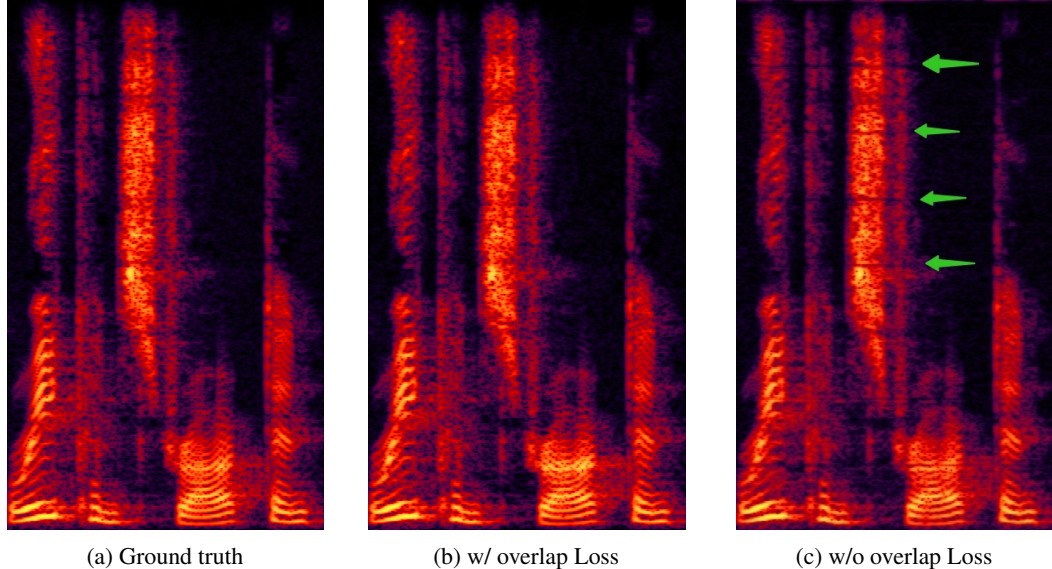

(a) Ground truth   (b) w/ overlap Loss   (c) w/o overlap Loss

Figure A.9: The effect of overlap loss. Omitting it results in a noticeable transition between subbands, as clearly illustrated by the green arrows.

## A.8 TABLES

Table A.2: Code repositories for model training.

| Model | repository |
|---|---|
| PriorGrad | https://github.com/microsoft/NeuralSpeech/tree/master/PriorGrad-vocoder |
| FreGrad | https://github.com/signofthefour/fregrad |
| Multi-Band Diffusion (MBD) | https://github.com/facebookresearch/audiocraft |
| EnCodec | https://github.com/facebookresearch/encodec |
| Vocos | https://github.com/gemelo-ai/vocos |
| BigVGAN | https://github.com/NVIDIA/BigVGAN |

Table A.3: Parameters for extracting Mel-spectrograms and complex spectrograms.

| Dataset | Sample rate (kHZ) | Window length | Hop length | FFT size | Mel bins |
|---|---|---|---|---|---|
| LJSpeech | 22.05 | 1024 | 256 | 1024 | 100 |
| LibriTTS | 24 | 1024 | 256 | 1024 | 100 |
| Opencpop | 44.1 | 2048 | 512 | 2048 | 100 |
| MTG-Jamendo | 44.1 | 2048 | 512 | 2048 | 100 |
| EnCodec Training | 24 | 1280 | 320 | 1280 | - |

Table A.4: Construction of the general Encodec test dataset

| Category | Subdirectory | Samples | Description |
|---|---|---|---|
| Speech | Expresso-test | 50 | From the Expresso dataset test set (Nguyen et al., 2023), randomly select 50 samples, segmenting any audio longer than 30 seconds into 10-second clips beforehand. |
| | HiFiTTS-test | 50 | Randomly select 50 samples from the test set of the Hi-Fi TTS dataset (Bakhturina et al., 2021). |
| | LibriTTS-test | 50 | Randomly select 50 samples from the test set of LibriTTS (Zen et al., 2019). |
| | Aishell3-test | 50 | Randomly select 50 samples from the test set of the Aishell3 dataset (Shi et al., 2020). |
| | JVS | 50 | Randomly select 50 samples from the entire JVS Corpus. (Takamichi et al., 2019). |
| | CML-TTS-test | 50 | Randomly select 50 samples from the test set of the CML-TTS dataset (Oliveira et al., 2023). |
| Vocal | Musdb-test-vocal | 60 | Segment the vocal track audio from the Musdb test set (Rafii et al., 2017) into 10-second clips, then randomly select 60 samples. |
| | CSD | 20 | Segment all audio in the CSD (Choi et al., 2020) into 10-second clips, then randomly select 20 samples. |
| | Opencpop | 20 | Randomly select 20 samples from the Opencpop dataset (Wang et al., 2022). |
| | ChineseOpera-monophonic | 60 | From the monophonic subset of the Chinese Opera Singing Dataset (Black et al., 2014), segment any audio longer than 30 seconds into 10-second clips, then randomly select 60 samples. |
| | JVS-Music | 60 | Randomly select 60 samples from the JVS-Music Corpus (Tamaru et al., 2020). |
| | RAVDESS | 60 | Randomly select 60 samples from the song subset of the RAVDESS corpus (Livingstone & Russo, 2018). |
| | Ccmusic-demo-vocal | 20 | From the demo audios of the Ccmusic Dataset (Liu & Li, 2021), select the vocal parts. Then, randomly select 20 samples from these for the test set, segmenting any audio longer than 30 seconds into 10-second clips beforehand. |
| Sound Effect | ESC-50 | 150 | Randomly select 150 samples from the ESC-50 (Piczak, 2015). |
| | Musdb-test-accompaniment | 40 | Segment the accompaniment track audio from the test set of the Musdb dataset (Rafii et al., 2017) into 10-second clips, then randomly select 40 samples from these clips. |
| | Musdb-test-mixture | 20 | Segment the mixture track audio from the test set of the Musdb dataset (Rafii et al., 2017) into 10-second clips, then randomly select 20 samples from these clips. |
| | ChineseOpera-polyphonic | 20 | From the polyphonic subset of the Chinese Opera Singing Dataset (Black et al., 2014), segment any audio longer than 30 seconds into 10-second clips, then randomly select 20 samples. |
| | OpenMIC-test | 40 | Randomly select 40 samples from the test set of OpenMix 2018 (Humphrey et al., 2018). |
| | Ccmusic-demo-music | 30 | From the demo audios of the Ccmusic Dataset (Liu & Li, 2021), select the non-vocal parts. Then, randomly select 20 samples from these for the test set, segmenting any audio longer than 30 seconds into 10-second clips beforehand. |

Table A.5: MOS and Objective evaluation metrics for RFWave, PriorGrad and FreGrad across various datasets.[6]

| Dataset | Model | MOS ↑ | PESQ ↑ | ViSQOL ↑ | V/UV F1 ↑ | Periodicity ↓ |
|---|---|---|---|---|---|---|
| LibriTTS | RFWave | $3.83_{\pm0.14}$ | 4.228 | 4.595 | 0.968 | 0.090 |
| | PriorGrad | $3.70_{\pm0.16}$ | 3.820 | 4.134 | 0.960 | 0.100 |
| | FreGrad | $3.68_{\pm0.15}$ | 3.758 | 4.278 | 0.960 | 0.099 |
| | Ground truth | $3.88_{\pm0.15}$ | - | - | - | - |
| Opencpop | RFWave | $4.26_{\pm0.17}$ | 4.176 | 4.564 | 0.990 | 0.049 |
| | PriorGrad | $3.79_{\pm0.21}$ | 3.404 | 4.512 | 0.988 | 0.064 |
| | FreGrad | $3.73_{\pm0.20}$ | 3.522 | 4.507 | 0.986 | 0.074 |
| | Ground truth | $4.33_{\pm0.18}$ | - | - | - | - |
| MTG-Jamendo | RFWave | $3.90_{\pm0.15}$ | - | 4.317 | - | - |
| | PriorGrad | $3.79_{\pm0.14}$ | - | 4.412 | - | - |
| | FreGrad | $1.89_{\pm0.17}$ | - | 3.765 | - | - |
| | Ground truth | $3.93_{\pm0.15}$ | - | - | - | - |

---

[6]The MOS of the ground truth on LibriTTS differs from that presented in Table 2 because these two MOS tests were conducted separately.

Table A.6: MOS and objective evaluation metrics for RFWave, EnCodec and MBD across various test sets.

| Test set | Bandwidth | Model | MOS | PESQ ↑ | ViSQOL ↑ | V/UV F1 ↑ | Periodicity↓ |
|---|---|---|---|---|---|---|---|
| Speech | 1.5 kbps | RFWave$_{(CFG2)}$ | 3.29±0.47 | 1.774 | 3.102 | 0.913 | 0.178 |
| | | EnCodec | 2.29±0.35 | 1.515 | 3.310 | 0.870 | 0.231 |
| | | MBD | 2.94±0.29 | 1.659 | 2.793 | 0.901 | 0.195 |
| | 3.0 kbps | RFWave$_{(CFG2)}$ | 3.85±0.39 | 2.421 | 3.582 | 0.935 | 0.137 |
| | | EnCodec | 2.85±0.47 | 1.967 | 3.746 | 0.919 | 0.166 |
| | | MBD | 2.88±0.33 | 2.194 | 3.219 | 0.915 | 0.166 |
| | 6.0 kbps | RFWave$_{(CFG2)}$ | 4.05±0.26 | 2.974 | 3.913 | 0.952 | 0.109 |
| | | EnCodec | 3.19±0.31 | 2.554 | 4.048 | 0.945 | 0.121 |
| | | MBD | 3.51±0.30 | 2.372 | 3.410 | 0.924 | 0.161 |
| | 12.0 kbps | RFWave$_{(CFG2)}$ | 3.96±0.22 | 3.393 | 4.158 | 0.965 | 0.089 |
| | | EnCodec | 3.52±0.22 | 3.104 | 4.250 | 0.961 | 0.095 |
| | | MBD | - | - | - | - | - |
| | Ground truth | | 4.13±0.23 | - | - | - | - |
| Vocal | 1.5 kbps | RFWave$_{(CFG2)}$ | 3.06±0.34 | 1.820 | 3.317 | 0.914 | 0.208 |
| | | EnCodec | 1.94±0.38 | 1.900 | 3.931 | 0.941 | 0.166 |
| | | MBD | 2.84±0.31 | 1.739 | 3.221 | 0.901 | 0.228 |
| | 3.0 kbps | RFWave$_{(CFG2)}$ | 3.47±0.34 | 2.467 | 3.793 | 0.942 | 0.152 |
| | | EnCodec | 2.63±0.37 | 1.900 | 3.931 | 0.941 | 0.166 |
| | | MBD | 3.26±0.45 | 2.426 | 3.655 | 0.929 | 0.175 |
| | 6.0 kbps | RFWave$_{(CFG2)}$ | 3.77±0.23 | 2.897 | 4.080 | 0.956 | 0.124 |
| | | EnCodec | 2.92±0.23 | 2.310 | 4.206 | 0.956 | 0.131 |
| | | MBD | 3.30±0.26 | 2.604 | 3.840 | 0.933 | 0.175 |
| | 12.0 kbps | RFWave$_{(CFG2)}$ | 3.83±0.31 | 3.147 | 4.273 | 0.964 | 0.109 |
| | | EnCodec | 3.40±0.29 | 2.679 | 4.373 | 0.965 | 0.114 |
| | | MBD | - | - | - | - | - |
| | Ground truth | | 4.27±0.24 | - | - | - | - |
| Soudn Effect | 1.5 kbps | RFWave$_{(CFG2)}$ | 3.24±0.43 | - | 2.906 | - | - |
| | | EnCodec | 2.62±0.42 | - | 3.314 | - | - |
| | | MBD | 3.33±0.36 | - | 2.932 | - | - |
| | 3.0 kbps | RFWave$_{(CFG2)}$ | 3.11±0.59 | - | 3.334 | - | - |
| | | EnCodec | 2.89±0.48 | - | 3.703 | - | - |
| | | MBD | 3.11±0.54 | - | 3.331 | - | - |
| | 6.0 kbps | RFWave$_{(CFG2)}$ | 3.27±0.27 | - | 3.682 | - | - |
| | | EnCodec | 3.17±0.24 | - | 4.020 | - | - |
| | | MBD | 3.44±0.23 | - | 3.496 | - | - |
| | 12.0 kbps | RFWave$_{(CFG2)}$ | 3.33±0.28 | - | 3.942 | - | - |
| | | EnCodec | 3.72±0.26 | - | 4.249 | - | - |
| | | MBD | - | - | - | - | - |
| | Ground truth | | 3.78±0.28 | - | - | - | - |

## A.9 ALGORITHMS

We present a simplified version of the pseudocode along with a comprehensive Python implementation for the two sampling algorithms. The detailed Python code delves into the intricacies of subband processing and inverse equalization, offering deeper insights into the algorithmic steps involved. Additionally, we provide a Python implementation for selecting time points of equal straightness.

### A.9.1 SAMPLING ALGORITHM

Following is the sampling algorithm ($X_t$ in time domain or frequency domain) with its pseudocode and Python implementation.

For simplicity, in the pseudocode, we've omitted the subband-related operations. Here, $X_t^{\mathfrak{f}}$ represents the combination of all $X_t^{i_{sb}}$ shown in Figure 1, and $v_t^{\mathfrak{f}}$ represents the combination of all $v_t^{i_{sb}}$ shown in Figure 1. For a comprehensive understanding of the processing steps, please refer to the Python implementation.

---

**Algorithm 1** Simplified Sampling Algorithm ($X_t$ in Time Domain)

---

**Require:**
1: $nn\_model$: pre-trained neural network model
2: $C$: conditional input
3: $\{t_i\}_{i=0}^N$: time steps where $0 = t_0 < t_1 < ... < t_N = 1$
**Ensure:**
4: Generated audio waveform $wave$
5:
6: $X_0 \sim \mathcal{N}(0,1)^{[1,T]}$  // Initialize with Gaussian noise
7: $X_t^{\mathsf{t}} \leftarrow X_0$
8:
9: **for** $i \leftarrow 0$ **to** $N-1$ **do**
10:  $dt \leftarrow t_{i+1} - t_i$  // Calculate step interval
11:  $X_t^{\mathfrak{f}} \leftarrow \text{STFT}(X_t^{\mathsf{t}})$  // Convert $X_t$ to frequency domain
12:  $v_t^{\mathfrak{f}} \leftarrow nn\_model.\text{predict}(X_t^{\mathfrak{f}}, t_i, C)$  // $v_t^{\mathfrak{f}}$ in frequency domain
13:  $v_t^{\mathsf{t}} \leftarrow \text{ISTFT}(v_t^{\mathfrak{f}})$  // Convert $v_t$ to time domain
14:  $X_t^{\mathsf{t}} \leftarrow X_t^{\mathsf{t}} + v_t^{\mathsf{t}} \cdot dt$  // Euler step in time domain
15: **end for**
16:
17: $X_1 \leftarrow X_t^{\mathsf{t}}$
18: $wave \leftarrow X_1$
19:
20: **return** $wave$

---

Detailed Algorithm 1: Sample Time Domain

```python
def sample_time_domain(model, mel, ts):
    '''
    :param model: A pre-trained time-domain RFWave model.
    :param mel: A batch of mel-spectrogram data, [batch_size, channels
    , num_frames].
    :param ts: The selected sampling time points.
    :return: The reconstructed audio waveform, [batch_size, num_frames
     * hop_length].
    '''

    batch_size, num_frames = mel.shape[0], mel.shape[2]
    noise_in_t = torch.randn([batch_size, model.hop_length *
    num_frames])
    # the STFT operation in Figure 1.
    noise_in_f = torch.stft(noise_in_t, **model.stft_kwargs)
    # Divide the noise spectrogram into subbands and reshape into the
    batch dimension
    noise_in_f = model.get_subband(noise_in_f)
    # repeat the mel-spectrogram to match the subbands accordingly.
    mel = torch.repeat_interleave(mel, model.num_bands, 0)

    z_in_t = noise_in_t
    z_in_f = noise_in_f
    vs = []   # used for selecting sampling time points
    for i in range(len(ts) - 1):
        t = ts[i]
        dt = ts[i + 1] - ts[i]
        # the model runs at frame-level feature.
        v_in_f = model.predict(z_in_f, t, mel)
        # place the subbands to construct a full band feature
        v_in_f = place_subband(model, v_in_f)
        # the ISTFT operation in Figure 1.
        v_in_t = torch.istft(v_in_f, **model.stft_kwargs)
        # the Euler step operates in the time-domain
        z_in_t = z_in_t + v_in_t * dt
        # the STFT operation in Figure 1.
        z_in_f = torch.stft(z_in_t, **model.stft_kwargs)
        z_in_f = get_subband(model, z_in_f)
        vs.append(v_in_t)
    # append the straight line velocity
    vs.append(z_in_t - noise_in_t)
    # inverse the waveform equalization.
    z_in_t = model.wave_equalizer.inverse(z_in_t)
    return z_in_t, vs
```

---

**Algorithm 2** Simplified Sampling Algorithm ($X_t$ in Frequency Domain)

---

**Require:**
1: $nn\_model$: pre-trained neural network model
2: $C$: conditional input
3: $\{t_i\}_{i=0}^N$: time steps where $0 = t_0 < t_1 < ... < t_N = 1$
**Ensure:**
4: Generated audio waveform $wave$
5:
6: $X_0 \sim \mathcal{N}(0,1)^{[d,F]}$     // Initialize with Gaussian noise
7: $X_t^{\mathfrak{f}} \leftarrow X_0$
8:
9: **for** $i \leftarrow 0$ **to** $N-1$ **do**
10:    $dt \leftarrow t_{i+1} - t_i$    // Calculate step interval
11:    $v_t^{\mathfrak{f}} \leftarrow nn\_model.\text{predict}(X_t^{\mathfrak{f}}, t_i, C)$
12:    $X_t^{\mathfrak{f}} \leftarrow X_t^{\mathfrak{f}} + v_t^{\mathfrak{f}} \cdot dt$    // Euler step
13: **end for**
14:
15: $X_1 \leftarrow X_t^{\mathfrak{f}}$
16: $wave \leftarrow \text{ISTFT}(X_1)$
17:
18: **return** $wave$

---

Detailed Algorithm 2: Sample Frequency Domain

```python
def sample_freq_domain(model, mel, ts):
    '''
    :param model: A pre-trained frequency-domain RFWave model.
    :param mel: A batch of mel-spectrogram data, [batch_size, channels
    , num_frames].
    :param ts: The selected sampling time points.
    :return: The reconstructed audio waveform, [batch_size, num_frames
     * hop_length].
    '''
    batch_size, num_frames = mel.shape[0], mel.shape[2]
    noise_in_f = torch.randn([batch_size, model.n_fft + 2,num_frames])
    noise_in_f = model.get_subband(noise_in_f)
    mel = torch.repeat_interleave(mel, model.num_bands, 0)

    z_in_f = noise_in_f
    vs = []  # used for selecting sampling time points
    for i in range(len(ts) - 1):
        t = ts[i]
        dt = ts[i + 1] - ts[i]
        # the model runs at frame-level feature.
        v_in_f = model.predict(z_in_f, t, mel)
        # the Euler step operates in the frequency-domain
        z_in_f = z_in_f + v_in_f * dt
        vs.append(v_in_f)
    # append the straight line velocity
    vs.append(z_in_f - noise_in_f)
    # place the subbands to construct a full band spectrogram
    z_in_f = model.place_subband(z_in_f)
    # inverse the stft normalization
    z_in_f = model.stft_normalizer.inverse(z_in_f)
    # convert the complex spectrogram to waveform
    z_in_t = torch.istft(z_in_f, **model.stft_kwargs)
    return z_in_t, vs
```

### A.9.2   SELECTING TIME POINTS OF EQUAL STRAIGHTNESS

Detailed Algorithm 3: Select Time Points

```python
def select_time_points(model, sample_fn, mel, N1=100, N2=10):
    '''
    This function is designed to select time points of equal
    straightness.
    :param model: A pre-trained RFWave model.
    :param sample_fn: sample_time_domain or sample_frequency_domain.
    :param mel: A batch of mel-spectrogram data, [batch_size, channels
    , num_frames].
    :param N1: The number of steps used for numerically estimating the
     straightness.
    :param N2: The number of steps for equal straightness
    :return:  A list of selected time points.
    '''

    # Step 1: Numerically estimate the trajectory
    eq_ts = torch.linspace(0, 1, N1 + 1)
    _, vs = sample_fn(model, mel, eq_ts)

    # Step 2: Calculate the deviation for each velocity step
    ds = []
    v_straight = vs[-1]
    for v in vs[:-1]:
        d = (v - v_straight).view(v.size(0), -1)
        d = torch.norm(d, p='fro', dim=1)
        ds.append(d.mean())

    # Step 3: calculate straightness
    # d_cum[-1] is the estimated straightness.
    d_cum = torch.cumsum(torch.stack(ds), dim=0)
    # s_inc is the increment of straightness for each step
    s_inc = d_cum[-1] / N2

    # Step 4: Select the time points of equal straightness
    ts = [0.]
    for i in range(1, N2):
        s_i = s_inc * i
        idx = torch.abs(d_cum - s_i).argmin()
        ts.append(idx / N1)
    return ts + [1.]
```

## A.10    SUBJECTIVE EVALUATION

We conducted the subjective listening test on our self-developed website platform. The platform allows users to play audio samples and rate the audio naturalness using a 5-point Mean Opinion Score (MOS) scale, where 1 indicates "poor/unnatural" and 5 indicates "excellent/natural".

For each subjective evaluation, we invited 30 listeners to participate. For each comparison experiment to be evaluated, we randomly selected 20-40 groups of audio samples from the corresponding test set for one listener to rate. Each group of audio includes ground truth audio, audio synthesized by RFwave, and audio synthesized by other models. The order of audio within each set was randomly shuffled.

All participating listeners were required to use headphones and give ratings from 1 to 5 based on the audio naturalness. The rating results were directly submitted to our evaluation website. Figure A.10 shows the interface of the evaluation website.

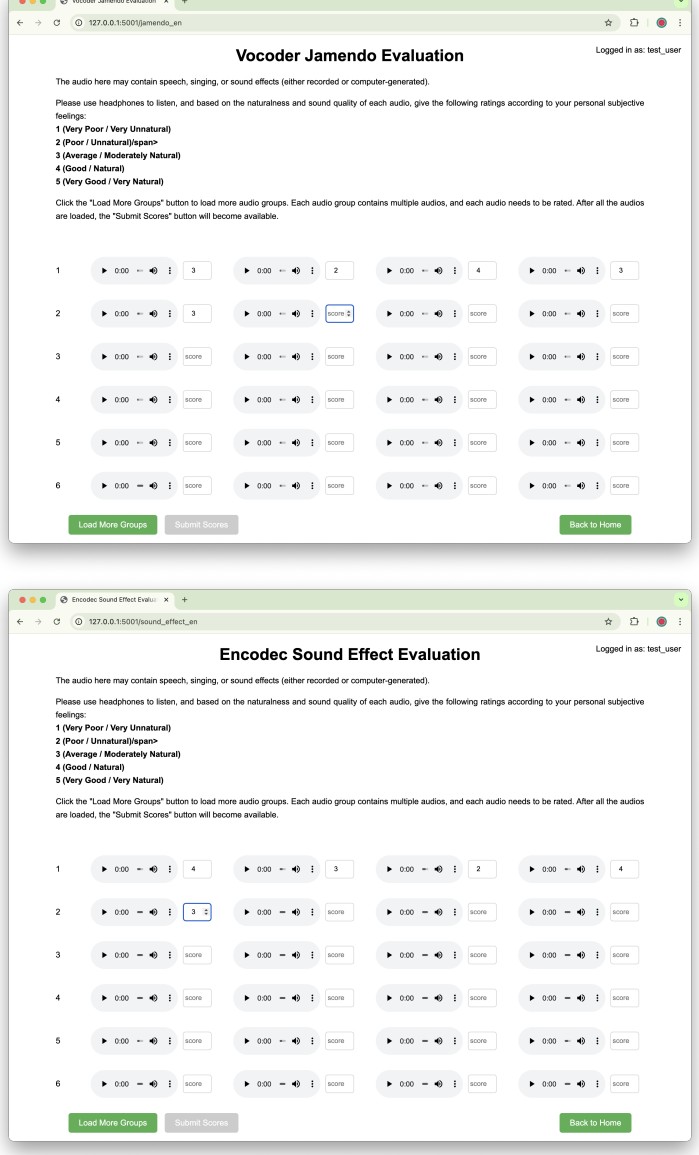

Figure A.10: Interface of our subjective evaluation platform.

