# OpenReview forum: "RFWave: Multi-band Rectified Flow for Audio Waveform Reconstruction"
_ICLR.cc/2025/Conference — ICLR 2025 Poster_

### Official Review · Reviewer_rq1Z · 2024-11-03

**Soundness:** 3
**Presentation:** 3
**Contribution:** 3
**Rating:** 8
**Confidence:** 4

**Summary:**

The present paper proposes RFWave, a rectified flow model for reconstructing audio waveforms from mel spectrograms or discrete tokens (EnCodec codes). The authors design their model to operate on slightly overlapping subbands, allowing for parallelized inference at greatly improved computational efficiency without introducing artifacts at the borders between subbands. The authors propose an energy-balanced loss and several auxiliary losses to enhance the output quality. The method shows highly competitive performance on the evaluated tasks compared to SOTA baselines, at an improved tradeoff between quality and inference speed.

**Strengths:**

The presented method is original in this application context, combining several existing works and ideas (rectified flow, ConvNeXt architecture, sub-band processing) with some problem-specific novel losses (energy-balanced loss, subband overlap loss) and a specialized time-point selection method for the ODE solver used for inference.

The evaluations show that the method offers a new exciting point along the tradeoff between inference speed and quality. It outperforms most baselines in audio quality and performs similarly to, or better than, BigVGAN while having only ~20% of the parameters, running more than twice as fast, and using ~40% less memory. The authors further conduct convincing ablation experiments on their proposed components.

The paper is overall structured well and written clearly, with both the clarity of the technical description and the use of language being good in general.

**Weaknesses:**

Some claims and described methodology lack clarity and should be expanded upon or rewritten more clearly.

* The description of the conducted subjective listening experiment is extremely terse with only a single sentence (lines 352-354). In my opinion, this is insufficient as several details here are key for the reproducibility and reliability of these results. It is necessary to add details on the used platform/service for evaluation, the number of listeners, the number and type of samples per listener, and ideally also the audio equipment used by listeners.

* Figure 1 is insufficient for understanding the model architecture. Specifically, I did not understand how the fusion of the $X_t$ and $C$ occurs before the linear block. The EnCodec bandwidth index $i_{bw}$ also occurs without further motivation or explanation here and is somewhat unclear.

* The relationship between the PQMF bank (lines 207-221) to the complex spectrogram feature-space representation (see abstract) is somewhat unclear. What is really used as input/output of each model variant?

* I appreciate that the authors provide an online demonstration, which is essential for the trustworthiness of the results. However, the selection of samples is suboptimal, particularly for speech where the ground truth signals are of low quality and affected by artifacts.
Please replace these or add new samples of higher ground-truth quality.

* The authors argue that using lower-frequency bands to predict higher-frequency bands can negatively impact the higher-frequency bands (lines 142-146), but later also argue that doing so can potentially increase consistency between bands (lines 280-281). Conditioning higher on lower bands is also not the same as processing all bands jointly (since there, also lower bands are estimated based on higher bands), so I do not fully follow the argument made here. The exact argument pro/contra processing low- and high-frequency bands jointly (or not) is somewhat unclear - please clarify.

* The authors seem to have convincing results for their "equal straightness" variant (Table 5), but somehow do not include these (at least) in Tables 6 and the comparison with diffusion-based methods, as they also note in the footnote on page 7. It is unclear to me why this is. Please update all metric results to present what the final RFWave variant is consistently, or list both the equal-straightness and non-equal-straightness variants in all relevant experiments for comparison.

* While the objective and subjective metrics speak for the method, from the online demo I find that the authors' method can introduce a kind of colorization and phasy artifacts that are not present in other methods, which are however not discussed in the paper. For transparency, could the authors add a short discussion of this limitation and potential reasons for it?

**Questions:**

* Were any methods trained on multiple types of audio (speech, music, sound) at the same time, or was a specialized model prepared of each RFWave and retrained baseline for *every* audio type? This is not clear to me from line 344, which would suggest that baselines were trained on multiple audio types at the same time, whereas Appendix A.2 suggests that RFWave was trained on every single dataset separately. If the authors compare dataset-specialized RFWave against multi-dataset baselines, such a comparison would be unfair.

* For completeness and better understanding, I would suggest adding a pointwise metric such as a simple L2 distance (or SNR in dB) of the reconstruction compared to the ground truth. I am asking since the presented metrics (MOS, PESQ, ViSQOL, F1, Periodicity) do not provide much insight in how the methods compare regarding pointwise distortion, but this is an important property, especially for the decoding task.

* The authors do not use the rectification step that a rectified flow model enables in principle, correct? Can they provide a reasoning for why they do not perform this step, which in theory should help improve the inference efficiency by further reducing the necessary number of sampling steps?

* Lines 47, 54-56, 70-71: It is a priori unclear why operating on STFT frames should be more efficient than operating on a waveform, since the STFT is redundant and introduces more data points to process. This seems to be more about frame-parallel processing of the DNN architecture rather than the STFT itself, could the authors clarify what they mean?

* In Section 3.3, what is meant precisely by timepoints being "chosen such that the increase in straightness is equal across each step"? I would have liked a more concrete description of the method used to obtain the time steps here - was this optimization-based, or somehow estimated in closed-form? What was the formal metric used to judge "equal straightness" across all intervals?

* What do the authors really mean by one model variant being capable of "operating in the time domain" (line 197-199)? In the abstract, the authors state that RFWave "generates complex spectrograms" -- in a flow/diffusion setting, shouldn't then the feature-space input also be a complex spectrogram? In this case, an STFT/iSTFT is used, and the ConvNeXt model performs 2D convolutions, and operates in the time-frequency domain, not in the time-domain, right? Please clarify this point here and also in the paper.

Minor points:

* In lines 204-206, the authors state that they interleave real and imaginary parts of each subband. Wouldn't this lead to a $2 d_s$-dimensional feature rather than a $d_s$-dimensional feature? (Same comment in line 227 where this notation is repeated)

* When the authors say "dimension-wise" (line 228), do they mean only $d_s$, or only $F$, or both $d_s$ and $F$?

* Can the authors add a reference for the claim that "speech energy decays exponentially with frequency, while music maintains a consistent distribution"? (lines 207-209)

* I do not follow the argument made in 3.2, lines 237-242, regarding MSE-based training. If "minor absolute distortions lead to significant relative error", shouldn't a model trained by minimizing MSE *avoid* such small absolute distortions rather than producing them? To me, the observation that the model exhibits residual noise sounds rather like a time-weighting or noise schedule issue than a problem with the MSE used within the training target?

* In the experimental setup section (4.1), I would suggest explicitly referring to Appendix A.1 which lists further training hyperparameters - I could not find such a reference and was missing these details in the main text before finding them in the appendix by chance.

* In the appendix, please provide (visually or quantitatively) the dynamic range used for producing each selection of shown spectrograms.

Typos:

* Lines 364 and 372: "Comparasion" -> "Comparison"
* Line 511: "lenght" -> "length"
* Table A.5: "PriroGrad" -> "PriorGrad"

---

> ### Author Response · Authors · 2024-11-22
> **Reply to Reviewer rq1Z (Part 1)**
>
> We sincerely appreciate the comprehensive review of our manuscript. We are particularly thankful for Reviewer rq1Z's keen interest and detailed inquiries, which have provided valuable insights for improving our work. Below are our thorough responses to each point raised.
>
> **W1: Details about the subjective listening experiment**
>
> Thank you for your inquiry. We have added the detailed information about the subjective evaluation to Appendix A.10 in the updated paper.
>
> **W2: The fusion of the $X_t$ and $C$ and the use of $i_{bw}$**
>
> Thank you for pointing this out. We have revised Figure 1 and modified Section 3.1 to improve clarity in the paper.
> - **Lines 186-187:** The $X_t^{i_{sb}}$, $C$, and Fourier features are concatenated along the channel dimension and
> then passed through a linear layer, forming the input that is fed into a series of ConvNeXtV2 blocks.
> - **Lines 189-190:** The $i_{bw}$ is utilized during the decoding of EnCodec tokens, enabling a single model to support EnCodec tokens with various bandwidths.
>
> **W3: The relationship between the PQMF bank and the complex spectrogram**
>
> In the time-domain model, PQMF is used exclusively for waveform equalization. We have revised lines 222-230 for clarity. It divides the full-band waveform into subbands, after which mean-variance normalization is applied to these subbands. The equalized subbands are then merged back into a single waveform, resulting in the equalized waveform $X_1$ which is used by the time-domain model.
>
> > **"What is really used as input/output of each model variant?"**
>
> We have clarified these aspects in the updated paper.
> - **Lines 194-199:** Our methodology offers two modeling options. The first involves mapping Gaussian noise to the waveform directly in the time domain, wherein $X_0$, $X_1$, $X_t$ and $v_t$ all reside in the time domain. The second option maps Gaussian noise to the complex spectrogram, placing  $X_0$, $X_1$, $X_t$ and $v_t$  in the frequency domain. Notably, $X_t^{i_{sb}}$ and $v_t^{i_{sb}}$ are consistently represented in the frequency domain, ensuring that the neural network runs at the frame level.
>
> **W4: New demo audios of higher ground-truth quality**
>
> Our Speech (LibriTTS) demo audio was randomly selected from [LibriTTS](https://openslr.org/60/)'s test-clean section. Since the LibriTTS dataset comes from the [LibriVox project](https://librivox.org/), its overall quality is poor, and many audio files inevitably are of low quality and affected by artifacts.
>
> Meanwhile, the speech portion of our general Encodec test dataset (Table A.4) contains cleaner speech. Therefore, we randomly selected new audio from the English Speech section of the general Encodec test dataset to create new demo pages. The webpage links are: [Comparision With Diffusion](https://rfwave-demo.github.io/rfwave/supplemental_codec_speech_testset_diffusion.html) , [Comparision With GAN](https://rfwave-demo.github.io/rfwave/supplemental_codec_speech_testset_gan.html) .
>
> **W5: Clarify using lower-frequency bands to predict higher-frequency bands**
>
> It is not contradictory that conditioning higher-frequency bands  on lower-frequency bands results in error accumulation while simultaneously enhancing the consistency among subbands. For instance, there exists a detrimental form of consistency since problems can occur when a higher-frequency band depends on a faulty lower-frequency band.
>
> >**"Conditioning higher on lower bands is also not the same as processing all bands jointly (since there, also lower bands are estimated based on higher bands)"**
>
> As noted in the MBD paper[1], "Training a diffusion model on full-band audio data would always provide the ground truth low frequencies when generating high frequencies. It ends up amplifying the errors committed at the beginning of the generation when unrolling the reverse process." Even when processing full-band audio, the lower bands inherently influence the determination of higher bands.
>
> Besides, if higher-band is conditioned on lower-band, they can not be predicted concurrently. This would leads to a latency propotional to the number of subbands. In our framework, we resort to overlap loss to maintain consistency during training and predict all subands concurrently during inference.
>
> [1] Robin San Roman, Yossi Adi, Antoine Deleforge, Romain Serizel, Gabriel Synnaeve, Alexandre Défossez: From Discrete Tokens to High-Fidelity Audio Using Multi-Band Diffusion. NeurIPS 2023

---

> ### Author Response · Authors · 2024-11-22
> **Reply to Reviewer rq1Z (Part 2)**
>
> **W6: Omission of equal straightness results in certain RFWave experiments (Table 1, Table 6 and Table 7)**
>
> The experiments in Tables 1 and 6 were conducted before we introduced  equal straightness. In Table 1, RFWave already achieves the best results without using equal straightness, and collecting Mean Opinion Scores (MOS) is both time-consuming and costly, which is why we did not reevaluate the model with equal straightness. In Table 6, models are evaluated using the same sampling method (except for DDMP, which uses its model-specific sampling method), ensuring a fair comparison. Regarding Table 7, the sampling time points for equal straightness are selected only once for a model, so it does not affect the sampling speed.
>
> **W7: Colorization and phasy artifacts**
>
> We are not entirely certain about the "colorization and phasy artifacts" referenced by the reviewer. It might be related to Figure `A.8(c)`. In this case, the model attempts to reconstruct a phase pattern in the silent background noise area where the phase is random. Adding the STFT loss and making the model overweight magnitude over phase slightly can alleviate this issue.
>
> **Q1: Concerns on dataset consistency for different models**
>
> In response to the concerns about dataset consistency, we have ensured that RFWave is compared to other models trained on consistent datasets. We acknowledge that the original description of the datasets may have led to some confusion, and we have clarified this in the updated paper.
>
> On page 7, lines 330-333, we explain our approach when benchmarking against diffusion vocoders. We trained separate models on the LibriTTS (speech), MTG-Jamendo (music), and Opencpop (vocal) datasets. Each model is tested on its respective dataset to ensure a comprehensive comparison across various audio categories.
>
> **Q2: Pointwise metric for the decoding task**
>
> We report the SI-SDR used in EnCodec paper.
> | Method   | 1.5 kbps | 3.0 kbps | 6.0 kbps| 12.0 kbps|
> |----------|----------|----------|----------|----------|
> | RFWave(CFG2) | 1.716   | 3.049   | 4.274   | 8.673   |
> | EnCodec  | -0.218  | 1.274   | 2.789   | 8.002   |
> | MBD     | -11.555 | -10.578 | -10.082 | -     |
>
> Our results show that RFWave consistently achieves better SI-SDR compared to EnCodec, especially when bandwidth is limited. Although MBD attains better perceptual quality than EnCodec, it appears not to effectively exploit the phase information encoded in EnCodec tokens.
>
> **Q3: Why not use the rectification step?**
>
> We chose not to use rectification because it introduces an additional training stage and requires a trade-off between performance and sampling speed. Our primary objective is to design a diffusion-type vocoder/decoder that has superior reconstruction quality and comparable speed to GANs, while also keeping the training process simple. This design approach increases the potential for wide adoption.
>
> Moreover, there are emerging new techniques [1] for high-quality one-step distillation. We plan to explore these in future research.
>
> [1] Cheng Lu, Yang Song: Simplifying, Stabilizing and Scaling Continuous-time Consistency Models.
>
> **Q4: Why operating on STFT frames should be more efficient than operating on a waveform?**
>
> Two key factors are crucial for overall computational efficiency: the **length** and **dimension (channels)** of the intermediate features that the model's backbone operates on.
>
> Utilizing complex STFT Spectrogram frames offers superior computational efficiency through a dramatic reduction in sequence length. Depending on the hop length configuration, the complex STFT Spectrogram can be 256 or 512 times shorter than the original waveform.
>
> Waveform-based diffusion models, such as those implemented in [diffwave](https://github.com/lmnt-com/diffwave/blob/master/src/diffwave/model.py#L154), [PriorGrad](https://github.com/microsoft/NeuralSpeech/blob/master/PriorGrad-vocoder/model.py#L171), and [FreGrad](https://github.com/kaistmm/fregrad/blob/main/model.py#L216), require the upsampling of conditional inputs (like Mel-spectrograms) to match the waveform length prior to processing. Consequently, all intermediate feature representations maintain the full length of the waveform throughout the network.
>
> In contrast, RFWave processes intermediate feature that match the length of the complex STFT spectrogram. This 256 or 512 times reduction in sequence length throughout the network significantly reduces computational overhead.
>
> While waveform-based diffusion models typically have intermediate feature dimensions of 32 or 64, RFWave model utilizes a dimension of 512. Despite the fact that RFWave employs 16x or 8x the feature dimensions, the substantial reduction in feature length ensures that the computational process remains significantly faster. This efficiency gain comes from the reduced sequence length, which outweighs the increase in feature dimensions, leading to enhanced computational efficiency overall.

---

> > ### Comment · Reviewer_rq1Z · 2024-11-25
> >
> > I'd like to thank the authors for addressing most of my comments adequately.
> >
> > Regarding my mentioned W7 and the authors' response, I would like to clarify. The issues on phasy artifacts and a kind of coloration that I perceive in demo page examples do not seem "related to Figure A.8(c)" to me. I apologize for using the wrong term "colorization" before – it should be "coloration". What I mean is a timbral distortion that does not match the ground-truth signal, see for instance [1*].
> >
> > As a concrete example, take the second audio example on [the newly provided clean speech demo page](https://rfwave-demo.github.io/rfwave/supplemental_codec_speech_testset_gan.html) – thank you for providing this. The word "children" sounds distorted in the RFWave reconstruction (spectrally modified, i.e. colored, especially in the mid-range), while the BigVGAN and Vocos reconstructions sound more natural. This becomes especially apparent when applying a bandpass filter between 2000 and 4000Hz (40dB rolloff, 3dB passband attenuation) to all methods' reconstructions and then listening. I hope this clarifies the effect I observed.
> >
> > **I was wondering if the authors can offer any explanation for this behavior since it seems to be present in many other examples as well.**
> >
> > [1*] Wältermann, Marcel. Dimension-based quality modeling of transmitted speech. Springer Science & Business Media, 2013.

---

> ### Author Response · Authors · 2024-11-22
> **Reply to Reviewer rq1Z (Part 3)**
>
> **Q5: A more concrete description of the equal straightness method**
>
> The time points were selected by numerical estimation. We add an algorithm for this in Appendix A.9.2.
>
>  **Q6 Clarify "operating in the time domain"**
>
>
> Thank you for raising these questions. We have revised Section 3.1 and Figure 1 of our paper to clarify the operational domains of our algorithm. Additionally, sampling algorithms for the two distinct approaches, one in the time domain and the other in the frequency domain, are provided in Appendix Section A.9.1. We hope these revisions address the reviewer's questions and provide a clearer understanding of our methodology.
>
> Our methodology offers two modeling options. The first involves mapping Gaussian noise to the waveform directly in the time domain, wherein $X_0$, $X_1$, $X_t$ and $v_t$ all reside in the time domain. The second option maps Gaussian noise to the complex spectrogram, placing  $X_0$, $X_1$, $X_t$ and $v_t$  in the frequency domain. Notably, $X_t^{i_{sb}}$ and $v_t^{i_{sb}}$ are consistently represented in the frequency domain, ensuring that the neural network runs at the frame level.
>
> Since our model is designed to function at the frame level, when $X_t$ and $v_t$ are in the time domain (specially, $X_1$ is the waveform and $X_0$ is noise of the identical shape, with $X_t$ and $v_t$ derived from Equation 3), the use of STFT and ISTFT, as illustrated in Figure 1, becomes necessary.
>
> When $X_t$ and $v_t$ reside in the frequency domain (i.e., $X_1$ is the waveform's complex spectrogram and $X_0$ is noise of the identical shape), STFT and ISTFT, as shown in Figure 1, are unnecessary.
>
> > **"In this case, an STFT/iSTFT is used, and the ConvNeXt model performs 2D convolutions, and operates in the time-frequency domain, not in the time-domain, right?"**
>
> In our framework, the ConvNeXtV2 block consistently performs 1D convolutions. The neural network process $X_t^{i_{sb}}$ of shape [batch_size, 2*$d_s$, num_frames].
>
> **Q7: $2d_s$-dimensional feature rather than a $d_s$-dimensional feature**
>
> The previous expression wasn't clear enough. Initially, when we stated that "$d_s$ denotes the dimension of a subband's complex spectrum," we intended to convey that the dimension includes the interleaved real and imaginary parts that form a complex spectrogram (not in the traditional complex data type sense). Thus, $d_s$ encompasses both the real and imaginary dimensions. Thank you for pointing this out. We have clarified the expression by changing "$d_s$ denotes the dimension of a subband's complex spectrum" to "$d_s$ represents the number of frequency bins in a subband's complex spectrum," and updated other instances in the text to $2d_s$ for greater clarity.
>
> **Q8:  "dimension-wise" (line 228) means only ds, or only F, or both ds and F?**
>
> It specifically refers to only $d_s$. More precisely, each frequency bin tracks its own mean and variance.
>
> **Q9: reference for the claim that "speech energy decays exponentially with frequency, while music maintains a consistent distribution"**
>
>  Thank you for pointing out this. We have included the reference in the updated version.
>
> [1] Jan Schnupp, Israel Nelken, and Andrew King: Auditory neuroscience: Making sense of sound. MIT press, 2011.
>
> **Q10: Argument made in 3.2, lines 237-242, regarding MSE-based training.**
>
> Thank you for pointing out this. We have revised this part for greater clarity.
>
> The MSE measures the absolute distortion between the predicted values and the ground truth. In silent regions, small absolute errors contribute minimally to the overall MSE loss, so the model does not prioritize eliminating them during training. This results in the model's inability to effectively suppress minor deviations in silent areas, potentially leading to perceptible noise. In contrast, larger errors in high-amplitude regions have a significant impact on the MSE loss, causing the model to focus more on reducing errors in these areas during training.
>
> **Q11: Referring to Appendix A.1 in the experimental setup section (4.1)**
>
> We have included the relevant details in the implementation paragraph of Section 4.1 in our original paper, which refers to Appendix A.1 for further information on the experimental setup.
>
>
> **Q12: Provide (visually or quantitatively) the dynamic range used for spectrograms**
>
> We used Adobe Audition for spectrogram visualization and the dynamic range is [-132, 0] dB for all the plots. We added this detail in the appendix.
>
> **We appreciate the attention to detail in identifying the typos. These have now been corrected.**

---

> ### Comment · Reviewer_rq1Z · 2024-11-25
>
> I'd like to thank the authors for their thorough explanations and modifications. I have only one main concern remaining here based on the answer provided to Q6.
>
> To me, the difference between Algorithm 1 (time-domain sampler) and Algorithm 2 (time-frequency domain sampler) seems too minor to plausibly explain why the time-frequency domain sampler performs significantly worse in Table 5 (row "frequency", see in particular the PESQ value). Thanks to the authors' explanations, I realize now that the network $nn_{model}$ always operates on complex spectrograms and outputs a velocity in the same (time-frequency) domain. The only difference, the added STFT and iSTFT, are both **linear** operators, so they should not make a meaningful difference for the update $v_t \cdot dt$ from each Euler step. The only difference is that the STFT is higher-dimensional than the time-domain signal, so not all possible complex spectrograms $X$ are consistent (valid STFTs), and hence one must in general project a complex spectrogram to the time domain and back to the time-frequency domain ($\text{STFT}(\text{iSTFT}(X))$) to ensure a complex spectrogram is consistent. All that Algorithm 1 seems to add is an implicit consistency projection on the predicted velocity $v_t$ by performing the iSTFT on it, but it is not clear why this is reasonable (the velocity $v_t$ does not itself correspond to a natural time-domain signal), and especially why it makes such an important quality difference. **I would appreciate if the authors could make any clarifying remarks regarding this or, if not possible, well-reasoned speculation.**
>
> **Note: I do not follow the authors' argument that "the ISTFT operation (Figure 1) introduces periodic signals that can enhance harmonic structures" and their claim that this is similar to the Snake activation** (see section 4.4 and Reply to Reviewer 5nT1). The iSTFT only creates meaningfully harmonic signals when there is organized sparsity in the frequency domain, but it is also perfectly capable of synthesizing white noise when the time-frequency coefficients are white noise (e.g. approximately for sibilant sounds), and does not meaningfully introduce harmonicity to the signal in this case.
>
> Additionally, I still find the statement (copied from the current paper version) that the authors' model "can operate with noisy sample in either the time or frequency domain" rather confusing. It suggests to me that either a larger part – e.g. the network architecture, feature extractor, etc. – must be modified to get a model that can operate in either of those domains or, alternatively, that the authors have made a novel and special modification to the network to allow for the network to accept both time and time-frequency representations. Only through the discussion and the authors' explanatory remarks here did I understand that the only modification is one made to the sampler.

---

> > ### Author Response · Authors · 2024-11-26
> >
> > **Q2:Superior performance of the time-domain model**
> >
> > While our paper only presents the sampling code, the training processes between the two approaches are also fundamentally different.
> >
> > For a simplified case, considering a specific complex spectrogram frame $S_1$, its noise $S_0$, and the model's output  $v_s(t)$. While the losses for frequency and time domain models can be formulated using the same equation in our paper (with different definitions of $X_1$ and $X_0$), their practical implementations differ:
> > - frequency domain loss: $L_{freq}=||(S_1 - S_0) - v_s(t)||^2$
> > - time domain loss: $L_{time}=||W^{IDFT}((S_1 - S_0) - v_s(t))||^2$
> >
> >
> > where $W^{IDFT}$ is inverse discrete fourier transform matrix[1].
> >
> > The time domain model demonstrates superior performance, which may attribute to $W^{IDFT}$ acts as an efficient weighting matrix that enhances model optimization. While the reviewer may overlook the training process, it's important to note that the other weights in the network adaptively co-evolve with $W^{IDFT}$ in the time domain model.
> >
> > [1] https://en.wikipedia.org/wiki/DFT_matrix
> >
> >
> > > I do not follow the authors' argument that "the ISTFT operation (Figure 1) introduces periodic signals that can enhance harmonic structures" and their claim that this is similar to the Snake activation
> >
> > We suggest that the $W^{IDFT}$ can be considered as a linear layer with fixed weights, and since these weights exhibit periodicity, it is reasonable to conclude that this transformation introduces periodicity to the final signal. While Snake introduces periodicity through activation function, our method achieves this through a fixed linear layer.
> >
> > However, it's important to note that the $W^{IDFT}$ should not be viewed in isolation. It is an integral part of the neural network that influences the learning and updates of other network weights during training. The generation of meaningful harmonic structures versus white noise is a result of the entire network's training process, not just the periodic properties of $W^{IDFT}$ alone. Similarly, a generator with Snake activation would also produce white noise without sufficient training, as the meaningful harmonic structure emerges through proper network optimization.
> >
> >
> > > It suggests to me that either a larger part – e.g. the network architecture, feature extractor, etc. – must be modified to get a model that can operate in either of those domains or, alternatively, that the authors have made a novel and special modification to the network to allow for the network to accept both time and time-frequency representations.
> >
> > Thank you for this insightful observation. You are absolutely correct. The differences between the time and frequency domain models extend well beyond the sampling method. While we focused on explaining the sampling method to facilitate understanding, it represents only one aspect of the architectural distinctions between these models.
> >
> > The fundamental differences encompass (as detailed in Section 3.1):
> >
> > - Network architecture adaptations (inclusion of STFT and ISTFT or not)
> > - Training process specifications (waveform equalization or STFT normalization)
> > - Loss function implementations - although these can be formulated using the same equation in our paper, their practical behaviors differ
> >
> > These comprehensive adaptations enable our models to effectively handle noisy samples in either time or frequency domains. We chose to highlight the sampling method as an accessible entry point for understanding the model variations, but this should not be interpreted as a comprehensive account of all architectural and operational differences between the time and frequency domain models.
> >
> >
> > Thank you for your careful and detailed review of our reply and paper content. Your feedback has helped us improve the paper and identify research issues. Thank you again for your enthusiasm. If you have any further questions, please feel free to ask.

---

> > ### Author Response · Authors · 2024-11-27
> >
> > We sincerely appreciate your detailed feedback. We have carefully addressed your comments and would welcome any additional thoughts or concerns you may have. Please feel free to reach out if further clarification is needed.

---

> > > ### Comment · Reviewer_rq1Z · 2024-11-27
> > >
> > > I appreciate the authors' detailed clarifications and their willingness to discuss, as well as the changes made to the paper. While I still do not fully agree with the arguments made regarding the DFT matrix inducing periodicity or harmonicity or regarding multi-band modeling effectively avoiding errors that single-band modeling does not, I believe these are rather minor points of contention.
> > >
> > > In my opinion, this is a very important paper for the field, particularly for closing the runtime performance gap between diffusion (and flow) methods and GAN-like methods. **The authors have convincingly shown that the introduced techniques work very well—to some extent contrary to intuition, making it all the more important that it is shown—and these techniques will be highly valuable for future follow-up works on flow models in audio processing. I am therefore increasing my score to 8.**
> > >
> > > A minor correction: According to the provided code, you do not use an iDFT matrix for the time-domain loss, but rather an iSTFT, which involves an overlapping sum of the result of multiple frame-wise iDFT and is thus different (mainly due to the overlap, which is related to the spectrogram consistency mentioned above). I believe the iDFT matrix somehow inducing a favorable weighting cannot fully explain the model variants' metric differences - it may instead lie in the implicit STFT consistency induced by going through iSTFTs, or potentially other reasons both the authors and I are overlooking—this could be a subject of future investigations.

---

> > > > ### Author Response · Authors · 2024-11-28
> > > >
> > > > Regarding the differences between frame-wise IDFT and ISTFT, you raise a valid point - we did not include overlap sum for a simplified explanation. Thank you for pointing out the role of overlap sum in the ISTFT operation as a potential factor enhancing spectrogram consistency. This perspective merits further investigation.
> > > >
> > > > Thank you for your thoughtful review and constructive feedback. We deeply appreciate your comments, which will undoubtedly help refine our understanding and improve our ongoing research. We are particularly grateful for your recognition of the paper's potential impact on the field.

---

> ### Author Response · Authors · 2024-11-26
>
> **Q1:The coloration artifacts**
>
> Thank you for bringing this important observation to our attention. While the occasional "coloration" artifacts have negligible perceptual impact, we acknowledge that this phenomenon warrants thorough analysis. We believe this issue may be caused by the inverse waveform equalization process. During training, the waveform equalization and inverse equalization process employs exponential moving average statistics that are continuously updated, with recent samples having larger influence. **The Mel-spectrogram and these statistics together determine the final energy of the PQMF subbands. This continuous updating makes it challenging for the model to learn consistent energy calibration for each subband from Mel-spectrogram input, as the statistics keep shifting.** During inference, the inverse operation employs frozen statistics. Although the exponential moving average statistics are representative of the training dataset, recent samples have stronger influence on the statistics, which may lead to minor mismatches between these statistics and the sample to be predicted. Considering the case the reviewer highlighted: the second PQMF subband (1500-3000 Hz) may be inversed with slightly mismatched statistics, resulting in marginally elevated energy levels that produce the artifacts the reviewer observed.
>
> A potential solution would be to freeze the statistics during training after they become sufficiently representative of the dataset. **With fixed statistics, the model can effectively calibrate the energy of the subbands based on the Mel-spectrogram input.** Even if these fixed statistics slightly deviate from the PQMF subbands, the model can adapt its weights to learn the correct energy mapping from the Mel-spectrogram input during training, provided that these statistics remain constant. However, when the statistics are continuously updated, they introduce additional variability in the final subbands' energy, making it more challenging for the network to learn consistent energy calibration from the Mel-spectrogram input.

---

### Official Review · Reviewer_kzjW · 2024-11-03

**Soundness:** 3
**Presentation:** 4
**Contribution:** 3
**Rating:** 8
**Confidence:** 4

**Summary:**

The paper introduces RFWave, a new method for reconstructing audio waveforms from Mel-spectrograms or discrete acoustic tokens. The proposed method operates on Mel-spectrogram frames and produces spectral subbands in parallel, boosting synthesis speed. Additionally, the paper presents several heuristics aimed at speeding up inference and improving quality, namely: scaling flow-matching loss based on frame energy, adding MSE loss on overlapping parts of predicted frames, incorporating STFT loss on the estimation 𝑋_1​ at time 𝑡, and selecting optimal points for the Euler method. The model consistently outperforms diffusion-based baselines and performs on par with GAN-based methods. The experiments demonstrate that RFWave achieves a good trade-off between generation speed, memory consumption, and sound quality.

**Strengths:**

- The paper is well written and easy to follow;
- The introduced heuristics are intuitive and reasonable;
- The experiments clearly demonstrate better performance of the proposed method compared to diffusion-based baselines and give good understanding of pros and cons compared to GAN-based ones.

**Weaknesses:**

- Interpretation of the experimental results aren’t always supported by the data provided in the corresponding tables.
For instance, 1) according to the authors, Table 3 demonstrates significant advantages over GAN-based methods but the confidence intervals overlap in all scores accept one (“Mixed”, which, however, may be the most important one). I think that a significance test may help to emphasize the superiority of your method. 2) the same problem with overlapping MOS intervals appears in Table 4. At least, MOS evaluation doesn’t support the claim that RFWave “excels” in all metrics being very close to MBD in the key subjective metric.
- Although Table 3 demonstrates better performance of RFWave on out-of-domain data, I still have doubts that such experimental setting  is practical: in a real-life scenario one would rather re-train a waveform generation model on the target domain (music in this case) than use a model trained on an out-of-domain data (speech). I would consider testing performance of BigVGAN fine-tuned on small amount of music data to get better understanding of advantages of RFWave brought by its better performance on out-of-domain data.
- Minor issues: “Tables 2” p.8:378, “velocity filed”, p.6:308

**Questions:**

- BigVGAN repository provides several versions of the model. Did you compare the performance of RFWave with the smaller ones in terms of sound quality and inference speed? Adding more flavors of BigVGAN into Table 7 would give better understanding of the strengths of your method.

---

> ### Author Response · Authors · 2024-11-22
> **Reply to Reviewer kzjW**
>
> We sincerely appreciate the reviewer's thorough evaluation and positive feedback. Below we address the concerns raised.
>
> **W1: Confidence intervals overlap in MOS in Table 3 and 5 and significance test**
>
> We observed that in other works, such as those involving BigVGAN (Table 2 in BigVGAN paper) and Vocos (Table 2 in Vocos paper), the MOS confidence intervals also exhibit overlaps. While this suggests that such overlaps are not uncommon in the field, to ensure rigor and address the reviewer's concerns, we have conducted significance tests and calculated the p-values.
>
> For Table 3 in our paper, we calculated the p-values for the MOS of RFWave compared to those of BigVGAN and Vocos for each category. The p-values indicate that RFWave generally outperforms Vocos across most categories. In comparison to BigVGAN, RFWave performs similarly in some categories but shows significant improvements in "other," "mixture," and the overall "average." This suggests that RFWave has better generalization capabilities when the auditory scene is more complex, as the "other" and "mixture" contain more diverse elements than a single instrument or vocal.
>
> **p-values for the MOS in Table 3**
>
> |                     | Category | BigVGAN | Vocos  |
> |---------------------|----------|---------|--------|
> | **RFWave**          | vocals   | 0.66281 | 0.00007|
> | **RFWave**          | drums    | 0.53152 | 0.09322|
> | **RFWave**          | bass     | 0.12080 | 0.00000|
> | **RFWave**          | other    | 0.01495 | 0.00000|
> | **RFWave**          | mixture  | 0.00000 | 0.00000|
> | **RFWave**          | average  | 0.00002 | 0.00000|
>
>
>
> For Table 4 in our paper, we calculated the p-values for the MOS of RFWave compared to those of MDB and EnCodec for each bitrate. RFWave demonstrates significant performance improvements over EnCodec  at 1.5, 3.0, and 6.0 kbps. It also shows significant advantages over MBD at 3.0 and 6.0 kbps.
>
> **p-values for the MOS in Table 4**
>
> |                     | Bitrate (kbps) | MBD     | Encodec |
> |---------------------|----------------|---------|---------|
> |   **RFWave**        | 1.5            | 0.28360 | 0.00000 |
> |   **RFWave**        | 3.0            | 0.00843 | 0.00007 |
> |   **RFWave**        | 6.0            | 0.03477 | 0.00000 |
> |   **RFWave**        | 12.0           | -       | 0.09570 |
>
>
>
> **W2: Is the out-of-domain experimental setting pratical, especially when models can be fine-tuned on target domain data?**
>
> In this evaluation, the purpose of assessing out-of-domain data is to evaluate the model's generalization capabilities. We followed the experimental setup used in the BigVGAN and Vocos papers, rather than suggesting that models trained on LibriTTS data should be deployed in music applications. We agree with the authors' viewpoint that "in a real-life scenario, one would rather re-train a waveform generation model on the target domain (music in this case) than use a model trained on out-of-domain data (speech)."
>
> The experiments demonstrate that RFWave exhibits better generalization capabilities compared to GAN-based models. Extending this to real-world products and applications, using RFWave offers advantages in the following scenarios:
>
> - When the generated audio involves a wide variety of sounds. As AI products evolve, modern smart speakers not only engage in conversations but also create music and various sound effects. However, it is challenging for training data to cover all possible audio types.
> - In cases where it is difficult to obtain training data for specific audio types due to privacy or other reasons, using a model trained on different audio data with RFWave can be a viable option.
>
> **Q1: Compare with more flavors of BigVGAN**
>
>
> To ensure comparability between our model and the BigVGAN and Vocos pre-trained models, we aligned the training datasets by using LibriTTS. The BigVGAN repository includes two models trained on LibriTTS: one with 112M parameters (`BigVGAN`) and another with 14M parameters (`BigVGAN-base`). We opted to use the `BigVGAN` model because the Vocos paper had already compared the `BigVGAN-base` with Vocos, revealing that while both produced similar audio quality, Vocos significantly outperformed the `BigVGAN-base` in terms of speed. Given the dual considerations of speed and audio quality, Vocos serves as a stronger baseline than the `BigVGAN-base`. Consequently, we chose Vocos as one of our baselines and did not include the `BigVGAN-base`.
>
> **Thank you for pointing out the typo in our paper. It has been corrected in the updated version.**

---

### Official Review · Reviewer_c6cp · 2024-11-03

**Soundness:** 2
**Presentation:** 2
**Contribution:** 2
**Rating:** 5
**Confidence:** 4

**Summary:**

This paper considers a method for reconstructing the waveform from Mel-spectrogram or discrete acoustic tokens. The method generates complex spectrograms and performs reconstruction on a frame level.  Reconstruction is performed using a straight transport trajectory, which is achieved in a few sampling steps. Audio generation can be achieved much faster than in real time.

**Strengths:**

Audio reconstruction from Mel-spectrogram is achieved with a few sampling steps, and audio is generated much faster than in real-time.

**Weaknesses:**

1. Algorithm Flow and Domain Clarity:
The paper suffers from readability issues, making it challenging to understand the overall flow of the proposed algorithm. Specifically, it is unclear whether the algorithm operates in the time domain, frequency domain, or both, as it seems to suggest that transitions between these domains are possible but lacks a clear explanation or visual aids (e.g., a flowchart). This lack of clarity weakens the reader's comprehension and hinders an appreciation of the technical merits of the algorithm. It would benefit the authors to provide a visual or detailed description to convey these aspects effectively.

2. Assumptions about Input Data (Mel-Spectrogram):
The paper assumes a mel-spectrogram input to produce waveform output. However, it does not clarify how this mel-spectrogram is initially obtained or what assumptions are made about the input data. Understanding the starting point and any assumptions about preprocessing steps is crucial to assessing the practicality of the proposed algorithm. The authors should explicitly state these assumptions and briefly explain how mel-spectrograms are typically derived in relevant application contexts.

3. Frequency and Time Domain Operations:
Whether the multiband rectified flow is used to reconstruct the complex spectrogram or the waveform directly remains ambiguous. While the paper mentions using the short-time Fourier Transform (STFT), which implies frequency-domain operations, it also suggests waveform reconstruction capabilities. This lack of specificity raises questions about whether the algorithm operates exclusively in one domain or both. A more precise discussion of domain-specific processes, including the role and interplay of STFT in this context, would enhance the reader’s understanding and the algorithm's technical robustness.

4. Role of ConvNeXt V2 and Learning Velocity:
The role of ConvNeXt V2 in learning the velocity field is unclear. Given that ConvNeXt V2 is a recent architecture capable of supervised and self-supervised learning, understanding how it contributes to the velocity-based learning within the algorithm is essential for evaluating the approach's novelty and effectiveness. A clear outline of ConvNeXt V2’s integration and function in this context, especially concerning velocity learning, would clarify the methodology and strengthen the reader's ability to assess its value.

5. Waveform Generation from Complex Spectrogram:
The method for generating the final waveform from a complex spectrogram is not well-explained. Although the paper references GAN-based methods, it lacks a detailed description of how this process is implemented in the proposed model. An explicit explanation of the techniques used to transition from the complex spectrogram to the waveform would clarify the reconstruction process, making it easier to evaluate the proposed method's strengths and limitations.

6. Independent Subband Reconstruction:
The decision to reconstruct subbands independently raises concerns about consistency across subbands. While the paper suggests that independent reconstruction mitigates error accumulation, it overlooks the strong correlations typically present among subbands, which, if neglected, may compromise audio quality. Discussing the potential drawbacks of independent subband reconstruction and any strategies for inter-subband consistency checks would address this issue, making the approach more comprehensive and practical.

7. Reconstruction Speed-Up Justification:
Although the algorithm claims to achieve a reconstruction speed over 100 times faster than real-time, it is unclear whether such a significant speed-up is necessary or beneficial for practical applications. While computational efficiency is generally advantageous, the paper could explore the trade-offs between speed, model complexity, and audio quality to justify the need for such extreme speed-ups in relevant application scenarios. A deeper analysis here could reinforce the practical relevance of the proposed method.

8. Novelty and Engineering-Based Solutions:
The paper's novelty is in question, as it mainly combines existing components (multiband rectified flow and ConvNeXt V2) rather than proposing a fundamentally new approach. While this engineering solution may have merit in specific applications, the lack of novel contributions limits the paper's impact. Strengthening the theoretical foundation or providing a unique methodological contribution would be beneficial in highlighting the algorithm’s distinct value.

In summary, the paper presents an engineering-focused approach for accelerated waveform reconstruction. Still, several aspects lack clarity, including domain transitions, assumptions about input data, the role of ConvNeXt V2, and consistency in subband reconstruction. Addressing these issues and offering more insight into the necessity of the speed-up and potential applications would significantly improve the paper's quality and technical rigor.

**Questions:**

See above

---

> ### Author Response · Authors · 2024-11-22
> **Reply to Reviewer c6cp (Part 1)**
>
> We greatly appreciate the thorough and meticulous review of our paper. We are especially grateful for Reviewer c6cp's strong interest in our work and the detailed questions raised, which will undoubtedly help us improve the manuscript. In the following, we provide our careful and comprehensive responses to each of the points raised.
>
> **W2: Assumptions about Input Data (Mel-Spectrogram)**
>
> Our model, designed for audio waveform reconstruction, transforms compressed, low-dimensional features back into perceivable audio. In practice, Mel-spectrograms and discrete token representations are commonly used as such compressed, low-dimensional features. Our baseline models, including BigVGAN, Vocos, PriorGrad, FreGrad, Encodec, and MBD, use these types of inputs (either Mel-spectrogram or Encodec tokens). Therefore, we adopted the same compressed, low-dimensional feature configurations in our experiments to facilitate a fair comparison and evaluate the performance and characteristics of our model.
>
> Regarding the configuration for extracting Mel-spectrograms, we have detailed this in Table A.3, which is consistent with the configuration used in the Vocos model. A more comprehensive introduction to mel-spectrograms is available in [the linked resource](https://huggingface.co/learn/audio-course/en/chapter1/audio_data#mel-spectrogram). In summary, the mel-spectrogram is one of the various compressed, low-dimensional features that our algorithm is capable of handling. Unlike Mel-spectrograms, complex spectrograms in the frequency domain can be losslessly converted to and from waveforms in the time domain using the ISTFT and STFT operations.
>
> **W1&W3&W5: The algorithm operates in the time domain, frequency domain, or both? Whether the multiband rectified flow is used to reconstruct the complex spectrogram or the waveform directly remains ambiguous. The method for generating the final waveform from a complex spectrogram is not well-explained.**
>
> We have revised Section 3.1 of our paper to clarify the operational domains of our algorithm. Additionally, sampling algorithms for the two distinct approaches, one in the time domain and the other in the frequency domain, are provided in Appendix Section A.9.1. These algorithms provide detailed steps on how the model's predicted velocity gradually transforms noise into target data, as well as the process of transforming a complex spectrogram into a waveform. We hope these revisions address the reviewer's questions and provide a clearer understanding of our methodology.
>
> Our methodology offers two modeling options. The first involves mapping Gaussian noise to the waveform directly in the time domain, wherein $X_0$, $X_1$, $X_t$ and $v_t$ all reside in the time domain. The second option maps Gaussian noise to the complex spectrogram, placing  $X_0$, $X_1$, $X_t$ and $v_t$  in the frequency domain. Whether $X_t$ is in the time domain or frequency domain, the input ($X_t^{i_{sb}}$ in Figure 1)  directly processed by the neural network is always in the frequency domain, meaning our neural network always operates at the frame level.
>
> Since our model is designed to function at the frame level, when $X_t$ and $v_t$ are in the time domain (specially, $X_1$ is the waveform and $X_0$ is noise of the identical shape, with $X_t$ and $v_t$ derived from Equation 3), the use of STFT and ISTFT, as illustrated in Figure 1, becomes necessary.
>
> When $X_t$ and $v_t$ reside in the frequency domain (i.e., $X_1$ is the waveform's complex spectrogram and $X_0$ is noise of the identical shape), STFT and ISTFT, as shown in Figure 1, are unnecessary.
>
> **W4: The role of ConvNeXt V2 in learning the velocity field is unclear.**
>
> In Section 4, we conducted an experiment, with results detailed in Table 6, where ResNet was used as a replacement for ConvNeXt V2. The findings indicate that ResNet resulted in inferior objective metrics and slower performance. Although ConvNeXt V2 serves as an effective backbone, its use is not the central novelty of our approach. We will provide further elaboration on this aspect in the final question.
>
> **W6: Reconstructing subbands independently raises concerns about consistency across subbands.**
>
> While independent parallel modeling of subbands can improve inference speed, mitigate error accumulation. However, it might introduce inconsistencies between adjacent frequency subbands. Therefore, we proposed an overlap loss. Inter-subband consistency is maintained through the proposed overlap loss, as detailed in Section 3.2. As each subband is predicted, the model ensures consistency between its overlapping section and the main section (as illustrated in Figure 2 of the paper). This overlapping region acts as an anchor, helping to preserve consistency across subbands.

---

> ### Author Response · Authors · 2024-11-22
> **Reply to Reviewer c6cp (Part 2)**
>
> **W7: It is unclear whether such a significant speed-up is necessary or beneficial for practical applications; the paper could explore the trade-offs between speed, model complexity, and audio quality**
>
> >**"It is unclear whether such a significant speed-up is necessary or beneficial for practical applications; "**
>
> Diffusion models are a key branch of generative models, widely used in text-to-image projects and important in waveform generation research. They offer greater training stability compared to GANs, avoiding issues like "posterior collapse" and "mode collapse". However, their slower speed limits their use as vocoders/decoders in real-time speech generation, where GANs are preferred. For example, projects like [ChaTTS](https://github.com/2noise/ChatTTS) use Vocos, [fish-speech](https://github.com/fishaudio/fish-speech) uses HiFi-GAN, [XTTSv2](https://github.com/coqui-ai/TTS) employs ParallelWaveGAN, and [BARK](https://github.com/suno-ai/bark) utilizes the EnCodec decoder. Diffusion-based methods are more common in music generation, where they excel over GANs due to less stringent real-time requirements, as seen in [audiocraft's](https://github.com/facebookresearch/audiocraft.git) use of a multi-band-diffusion (MBD) decoder.
>
> Achieving a diffusion-based vocoder/decoder with a latency lower than 0.01 (time consumed/duration of generated audio) that is comparable to GAN while maintaining high-quality generated audio is meaningful for the community. This advancement addresses a critical bottleneck and could potentially broaden the applicability of diffusion vocoder/decoder in real-time applications.
>
> Additionally, compared to previous diffusion models, our model significantly reduces computational costs, which in turn decreases energy consumption.
>
> > **"the paper could explore the trade-offs between speed, model complexity, and audio quality"**
>
> In Section 4, we conducted experiments with models of half and double the original size, with results detailed in Table 6. Our findings indicate that increasing the backbone size enhances performance but reduces efficiency.

---

> ### Author Response · Authors · 2024-11-22
> **Reply to Reviewer c6cp (Part 3)**
>
> **W8: The paper's novelty is in question, as it mainly combines existing components**
>
>
>
> Although GAN-based methods require complex discriminator design and suffer from mode collapse issues, they remain the mainstream approach in Audio Reconstruction. While diffusion-based models have overcome these problems, they have not yet become mainstream, primarily due to their significantly slower inference speed compared to GAN-based methods.
>
> Our model addresses this by operating on frame-level features and utilizing Rectified Flow to reduce the number of sampling steps during generation. Additionally, it predicts multi-band audio in parallel. These innovations significantly enhance the generation speed of diffusion-based models, improving inference speed by an order of magnitude compared to original diffusion-based models, and closing the gap with GAN-based models. By doing so, we address the major limitation that has hindered the widespread application of diffusion-based models for waveform reconstruction.
>
> This paper introduces several novel elements:
>
> - **Diffusion model operating on frame-level feature to match the speed of GAN：** RFWave operates on frame-level complex STFT spectrogram features, achieving superior computational efficiency by dramatically reducing the sequence length—256 or 512 times shorter than the original waveform, depending on the hop length configuration. Compared to diffusion models that process features at the full waveform length, RFWave significantly reduces computational overhead. To our knowledge, this is the first instance where a diffusion vocoder/decoder achieves more than 100x real-time speed. This advancement addresses a critical bottleneck and could potentially broaden the applicability of diffusion vocoders/decoders in real-time applications. (A more detailed explanation of the speed-up is provided in response to Q2.)
>
>
> - **Energy-balanced loss for Rectified Flow training**: The mean squared error (MSE) measures the absolute distortion between predicted values and the ground truth. In silent regions, small absolute errors have minimal impact on the overall MSE loss, which means the model may not focus on eliminating them during training. This can lead to the model's inability to effectively suppress minor deviations in silent areas, potentially resulting in perceptible noise. We introduce an energy-balanced loss to address this issue, ensuring that such areas receive appropriate weight during training.
>
> - **Overlap loss to maintain consistency among subbands**: While parallel multi-band prediction can prevent error accumulation and enhance inference speed, it may cause inconsistencies between frequency bands. We propose the use of overlap loss to resolve these inconsistencies, ensuring coherent audio output across subbands.
>
> - **Equal straightness to enhance sample quality for free**: Equal straightness ensures that the difficulty of each Euler step remains consistent, requiring the model to take more steps in more challenging regions. By using the same number of sampling steps, this method outperforms the equal interval approach, enhancing sample quality without additional computational cost.

---

> ### Author Response · Authors · 2024-11-27
>
> Dear Reviewer c6cp,
>
> Thank you for your valuable comments and questions. We have carefully implemented all the corresponding revisions to the paper as detailed in our previous reply messages. As we are approaching the revision deadline, we would greatly appreciate your feedback on whether the revised version adequately addresses your concerns.
>
> Best regards,
>
> RFWave Authors

---

> ### Author Response · Authors · 2024-11-28
>
> Dear Reviewer c6cp,
>
> Thank you for your thoughtful feedback and questions. We hope that our responses and updates to the paper have addressed your concerns. If you have any further questions, please feel free to let us know. We are always looking forward to hearing your feedback.

---

> ### Author Response · Authors · 2024-11-30
>
> Dear Reviewer c6cp,
>
> Thank you again for your valuable comments during the review process. We hope that our previous responses have clarified your concerns. If you have any further questions or suggestions, please feel free to share them with us. We greatly appreciate your time and effort.
>
> Best regards,
> Authors

---

### Official Review · Reviewer_5nT1 · 2024-11-04

**Soundness:** 2
**Presentation:** 3
**Contribution:** 2
**Rating:** 5
**Confidence:** 4

**Summary:**

An efficient diffusion-based vocoder is proposed. The strengths of this paper are claimed to be that it generates the complex spectrogram of STFT at the frame level and that it can be generated in 10 denoising steps.

**Strengths:**

- Experimental results have demonstrated superior sound quality across multiple metrics compared to conventional methods.
- The strength of this paper appears to lie in the application of the latest technique, rectified flow, which enables efficient sampling. This aspect of the work stands out as a significant contribution.
- This work introduces three additional loss functions to improve performance. The additional loss functions appear to consistently improve performance.

**Weaknesses:**

- The task addressed in this paper is rather conventional, using standard datasets and common evaluation methods. As a result, the work comes across as a minor variation of existing research. While this is certainly important work, it lacks a clear element of originality. It is unfortunate that, in a field where many similar approaches already exist, this work appears to be another study that only slightly improves existing benchmarks.
- Moreover, the approach is not theoretically groundbreaking either. At first glance of the contribution part, the proposed method appears to be a combination of existing ideas, making its originality and novelty less immediately clear.
- Some spectrograms are shown, but the differences are not immediately apparent, making it unclear what aspects should be evaluated. It would be helpful to provide a little more explanation rather than leaving the interpretation up to the reader saying see the high-frequency components.

**Questions:**

- I find it difficult to understand the rationale behind dividing into subbands when working in the STFT domain. If we are considering the STFT, it seems that the signal is already inherently divided into frequency components.
  - Also, this paper mentions introducing a loss related to overlap in order to mitigate inconsistencies caused by subband division. However, inconsistencies between subbands are not limited to adjacent bands but may also impact the coherence of harmonic components. This could also be considered a potential drawback of subband division.
  - The paper claims improved performance on harmonics compared to conventional methods with some empirical examples, but it is not clearly justified why it is effective.

- The STFT introduces redundancy in the representation due to overlapping frames compared to the original waveform. Consequently, the assertion that complex STFT spectrograms could be generated much faster than waveforms is not immediately intuitive. A more detailed and clearer explanation is required to substantiate this claim.

---

> ### Author Response · Authors · 2024-11-22
> **Reply to Reviewer 5nT1 (Part 1)**
>
> We sincerely thank reviewer 5nT1 for the thorough review and valuable constructive feedback. We address each of the concerns in detail below.
>
> **W1: The task addressed in this paper is rather conventional.**
>
> Audio waveform reconstruction is still not a fully solved problem. Unlike GAN-based methods, diffusion-based approaches neither require the design of complex discriminators nor suffer from mode collapse issues. However, despite the existence of works like Diffwave and PriorGrad, diffusion-based methods have not yet become mainstream in the waveform reconstruction, unlike their success in text-to-image generation. The primary reason for this is their slower inference speed compared to GAN-based methods.
>
> Our model addresses this by operating on frame-level features and utilizing Rectified Flow to reduce the number of sampling steps during generation. Additionally, it predicts multi-band audio in parallel. These innovations significantly enhance the generation speed of diffusion-based models, improving inference speed by an order of magnitude compared to original diffusion-based models, and closing the gap with GAN-based models. By doing so, we address the major limitation that has hindered the widespread application of diffusion-based models for waveform reconstruction.
>
> **W2: Not theoretically groundbreaking, originality and novelty less immediately clear**
>
> This paper introduces several novel elements:
>
> - **Diffusion model operating on frame-level feature to match the speed of GAN：** RFWave operates on frame-level complex STFT spectrogram features, achieving superior computational efficiency by dramatically reducing the sequence length—256 or 512 times shorter than the original waveform, depending on the hop length configuration. Compared to diffusion models that process features at the full waveform length, RFWave significantly reduces computational overhead. To our knowledge, this is the first instance where a diffusion vocoder/decoder achieves more than 100x real-time speed. This advancement addresses a critical bottleneck and could potentially broaden the applicability of diffusion vocoders/decoders in real-time applications. (A more detailed explanation of the speed-up is provided in response to Q2.)
>
>
> - **Energy-balanced loss for Rectified Flow training**: The mean squared error (MSE) measures the absolute distortion between predicted values and the ground truth. In silent regions, small absolute errors have minimal impact on the overall MSE loss, which means the model may not focus on eliminating them during training. This can lead to the model's inability to effectively suppress minor deviations in silent areas, potentially resulting in perceptible noise. We introduce an energy-balanced loss to address this issue, ensuring that such areas receive appropriate weight during training.
>
> - **Overlap loss to maintain consistency among subbands**: While parallel multi-band prediction can prevent error accumulation and enhance inference speed, it may cause inconsistencies between frequency bands. We propose the use of overlap loss to resolve these inconsistencies, ensuring coherent audio output across subbands.
>
> - **Equal straightness to enhance sample quality for free**: Equal straightness ensures that the difficulty of each Euler step remains consistent, requiring the model to take more steps in more challenging regions. By using the same number of sampling steps, this method outperforms the equal interval approach, enhancing sample quality without additional computational cost.
>
> **W3: Differences in spectrograms are not immediately apparent**
>
> We have provided a description for each plot. However, it's possible that the descriptions were placed somewhat distant from their corresponding figures, which might have led to confusion. To address this issue, we have added more detailed information directly in the title of each spectrogram and highlighted the differences within the spectrograms themselves.

---

> ### Author Response · Authors · 2024-11-22
> **Reply to Reviewer 5nT1 (Part 2)**
>
> **Q1: Dividing into subbands when working in the STFT domain**
>
> Regarding this, on page 4, line 208-209, we extract the subbands by equally dividing the full-band complex spectrogram. The model then processes these subbands as independent samples.
>
> **Q1.1: Inconsistencies between subbands are not limited to adjacent bands**
>
> In Section 3.2, the overlap loss is designed to ensure consistency between consecutive subbands, such as aligning subband 1 with subband 2, subband 2 with subband 3, and so forth. By establishing consistency between adjacent subbands—such that if subband 1 aligns with subband 2, and subband 2 aligns with subband 3, then subband 1 inherently aligns with subband 3—we aim to achieve global coherence throughout the entire spectrogram, including its harmonic components. This strategy effectively mitigates the potential drawbacks of subband division by preserving the integrity of the overall audio signal.
>
> **Q1.2: Reasons for imporved performance on harmonics**
>
> The effectiveness of our approach in improving harmonics can be attributed to two key aspects:
>
> 1. **Rectified Flow**: This method has demonstrated strong generative modeling and generalization capabilities, which contribute to enhanced performance on harmonic content.
>
> 2. **ISTFT Operation**: In the time-domain model, the ISTFT operation (Figure 1) introduces periodic signals that can enhance harmonic structures. The inverse Discrete Fourier Transform (IDFT) matrix is composed of sinusoidal functions with frequencies [0, 1/N, ..., (N − 1)/N], where N represents the number of FFT points. This mechanism is akin to the Snake activation function used in BigVGAN, which also incorporates periodic signals within the generator.
>
> **Q2: Given the redundancy introduced by STFT, why is RFWave able to achieve faster generation?**
>
> Two key factors are crucial for overall computational efficiency: the **length** and **dimension (channels)** of the intermediate features that the model's backbone operates on.
>
> Utilizing complex STFT Spectrogram frames offers superior computational efficiency through a dramatic reduction in sequence length. Depending on the hop length configuration, the STFT Spectrogram can be 256 or 512 times shorter than the original waveform.
>
> Waveform-based diffusion models, such as those implemented in [diffwave](https://github.com/lmnt-com/diffwave/blob/master/src/diffwave/model.py#L154), [PriorGrad](https://github.com/microsoft/NeuralSpeech/blob/master/PriorGrad-vocoder/model.py#L171), and [FreGrad](https://github.com/kaistmm/fregrad/blob/main/model.py#L216), require the upsampling of conditional inputs (like Mel-spectrograms) to match the waveform length prior to processing. Consequently, all intermediate feature representations maintain the full length of the waveform throughout the network.
>
> In contrast, RFWave processes intermediate feature that match the length of the complex STFT spectrogram. This 256 or 512 times reduction in sequence length throughout the network significantly reduces computational overhead.
>
> While waveform-based diffusion models typically have intermediate feature dimensions of 32 or 64, RFWave model utilizes a dimension of 512. Despite the fact that RFWave employs 16x or 8x the feature dimensions, the substantial reduction in feature length ensures that the computational process remains significantly faster. This efficiency gain comes from the reduced sequence length, which outweighs the increase in feature dimensions, leading to enhanced computational efficiency overall.

---

> ### Author Response · Authors · 2024-11-28
>
> Dear Reviewer 5nT1,
>
> Thank you for your valuable feedback, which is crucial for improving our research and presentation. We welcome any additional suggestions or concerns you may have.
>
> Additionally, we would like to highlight that our method is the first diffusion-type approach capable of achieving audio generation speeds comparable to GAN-based models while maintaining exceptionally high audio quality. It is also at least tens of times faster than other existing diffusion-based models.
>
> We have carefully addressed your previous comments. If you need any further clarification, please feel free to reach out.

---

> ### Author Response · Authors · 2024-11-30
>
> Dear Reviewer 5nT1,
>
> Thank you for your comments during the review process. We hope that our previous responses have addressed your concerns. If you have any further questions or suggestions, please feel free to let us know. Your feedback is important to us.
>
> Best regards,
> Authors

---

> > ### Comment · Reviewer_5nT1 · 2024-12-02
> > **Response by Reviewer**
> >
> > Apologies for the delay in responding to your comments. I would like to address a few points.
> >
> > **W3** Upon reviewing the figures again, I cannot clearly recall the previous state, but it seems that the discussion on harmonics has become easier to understand compared to before.
> >
> > **Q2**
> > Thank you for your comment. I fully agree with your observation that increasing the hop size (e.g., from sample-wise synthesis, as in WaveNet, to 256 or 512 samples) effectively reduces the sequence length.
> >
> > That said, my question focuses on the rationale for applying the Fourier transform to each frame in this context. For instance, methods like WaveGlow, as I recall, achieved sequence length reduction by grouping chunks of 4 or 8 samples without relying on a Fourier transform to generate complex spectra. (Since FFT is a full-rank linear transformation, it should theoretically preserve all the information from the waveform, meaning that either representation should be equally valid in principle. Of course, there are likely differences in how easily the model can learn each representation.)
> >
> > Therefore, it seems that the reduction in sequence length is primarily due to increasing the hop size rather than the use of the STFT itself. Clarifying this distinction would address my concern and help resolve ambiguity.

---

> > > ### Author Response · Authors · 2024-12-02
> > >
> > > Thank you for your insightful comments. I would like to clarify that in our framework, the sequence length reduction by hop size is specifically a result of applying STFT to the waveform. This is why we emphasize in our paper that model operations on frame-level features can improve processing speed.
> > >
> > > You raise a valid point about sequence length reduction being achievable through various methods, such as sample grouping in WaveGlow without using Fourier transform. As you correctly noted, "there are likely differences in how easily the model can learn each representation." Indeed, this observation aligns with our choice of STFT, as high-frequency information in raw data is typically difficult for neural networks to learn directly [1]. By transforming to the frequency domain, we can represent these high-frequency components as distinct frequency bins, making them more manageable for the model to process.
> > >
> > > We will add a footnote on page 3, line 153 (where we discuss that 'the neural network consistently operates at the STFT frame level') to clarify that the reduction in sequence length is achieved through windowing waveform sample points when applying STFT.
> > >
> > > Thank you again for your valuable feedback. We hope our clarifications will be considered in your final assessment. Please don't hesitate to reach out if additional clarification is needed.
> > >
> > > [1] Tancik et al., "Fourier Features Let Networks Learn High Frequency Functions in Low Dimensional Domains"

---

### Official Review · Reviewer_2uM7 · 2024-11-05

**Soundness:** 4
**Presentation:** 3
**Contribution:** 3
**Rating:** 8
**Confidence:** 3

**Summary:**

In this paper the authors propose a new model called RFWave (Rectified Flow Wave) for achieving audio reconstruction from Mel Spectrograms or Encodec tokens. They propose training an efficient ConvNeXtV2
 backbone (a state-of-the-art convolutional architecture) using the rectified flow (diffusion) paradigm, that operate on STFT spectrograms bands for improved efficiency (than training directly in waveform domain). Each spectrogram band is processed independently by the network, and are concatenated before the inverse STFT (differently than the previous Multi-band Diffusion conditional approach of Roman et al. 2023, in which error accumulates from low to high frequency bands). The model can be trained both by performing the noising in time-domain, where is data is mapped first to STFT and back via the iSTFT before applying the loss, or trained
directly in STFT domain (authors show improved results on time-domain nosing approach). Methodologically the authors propose three new losses and new sampling algorithm for rectified flow. The losses are: a weighted loss which scales the STFT band by the variance of the ground truth velocity band (which is proportional to its energy), such that nearly-silent portions of data do not result in noisy outputs, an overlap loss which equalises
the outputs of different bands at their boundaries and an STFT loss computed via the rectified flow prediction at time $t$. The novel sampling algorithm choses sampling times by subdividing the flow trajectory in intervals
 of equal straightness (the integral of the difference between the ground truth velocity direction and the predicted velocity). The authors perform a solid experimental evaluation, comparing both with recent diffusion-based models and SOTA GAN-based models, showcasing very good results.

**Strengths:**

- While methodologically the authors borrow many ideas from previous papers, e.g., the model architecture ConvNeXtV2 is not a novelty of the paper, and training in the STFT domain is a classic choice in audio deep learning, it is very interesting how the combination of those ideas can bring us closer on surpassing the fundamental limitations in efficiency of diffusion models from a practical point of view. Seeing in Table 7 that actually performing 10 inference step of RFWave we have a way higher realtime factor than a BigVGAN (while having similar quality performance on LibriTTS and improved OOD performance on MUSDB18) is a strong indication that the inference time gap between diffusion models and GAN models is finally disappearing. And the best thing here is that one obtains this reduction without having to resort to distillation techniques (which are notably difficult to perform, especially consistency distillation), but only resorting to reasonable design choices, especially the choice of the architecture.
- The results in the experimental section are very good for the proposed model, putting it as a novel strong baseline in the audio reconstruction setting. Additionally, the ablations in Tables 5 and 6 give a detailed view on why the design choices in the paper contribute to the overall performance.
- The proposed losses and the idea of equal straightness intervals are valuable not only in the audio reconstruction context but also more generally when dealing with other types of STFT based models / other generative settings for rectified flow models.

**Weaknesses:**

- The paper in some parts is not very clear in its explanations. For example in the Energy-balanced Loss part of Section 3.2 (Loss Functions), from first (and repeated) readings, I could not understand what was the  problem with the MSE loss. After different readings I interpreted it in the following way: small absolute errors in silent regions contribute little to the overall MSE loss, so the model doesn't prioritize eliminating them. This means the model doesn't effectively suppress minor deviations in silent areas, resulting in perceptible noise. Thus larger errors in high-amplitude regions significantly impact the MSE loss, causing the model to focus more on reducing those errors during training. Another part that was not clear was all Section 3.3 (Selecting Time Points for Euler Method). Here, first, straightness should be defined for a time $s$, as $S(v, s)$, setting the definite integral from $0$ to $s$ (using $s$ to leave the $t$ in $dt$). I believe authors use this because they took it from the original paper on rectified flow. Then writing “allowing to take more steps in more challenging regions” is pretty confusing. The term “allowing” is not ideal, I would use “requiring”. I understood the concept thinking that if in an interval the straightness does not change much, it means that we can take bigger steps, saving on the overall number of steps. I please ask the authors to re-write these parts in a more understandable way.
- While the experimental sections is done in a good way as mentioned in the Strengths part, the fact that the authors trained RFWave on their proposed large-scale dataset (combining Common Voice 7.0, DNS Challenge 4, MTG-Jamendo, FSD50K and AudioSet) and that Vocos, BigVGAN, EnCodec, and MBD are evaluated using the public pre-trained models is somehow problematic: it could be that Vocos, BigVGAN, EnCodec, and MBD are under-trained with respect to the authors dataset. I would like the author to comment on this and at least show that the training sources are comparable.
- It would have been interesting to compare also with latent diffusion models, where the network operates in another domain that is more compressed as the waveform domain, maybe a fine-tuned version of Stable Audio Open, given their widespread adoption.

**Questions:**

Here I list questions and typos:
- Line 174: “can be” to “can be an”
- Line 184:  Fourier features of what? Of the noisy sample? How we concatenate the conditional inputs? They share same dimensionality?
- Line 205: If interleaved then shouldn’t be $2d_s$?
- Line 219: “For waveform equalization… computed during training” This part as well refers to PQMF? Or is distinct?
- Line 228: “Subsequent processing involves the dimension-wise mean-variance normalization of the complex spectrogram.” As well here there is the mean-variance normalization as in time domain?
- Line 246: Why compute the standard deviation and not the energy altogether? Is there any reason for this?
- Line 252: Space after the $\min$.
- Line 295: Isn’t $X_0$ the data point and $X_1$ the noisy point? So shouldn’t we approximate  the STFT Loss using $X_0$?
- Line 307: “filed” is “field”.  In the integral use $\mathbb{E}$. Also I think the straightness should have argument a time $s$ (see Weakness section).
- Line 382: Why snake activation enhances out-of-domain data generation capabilities?
- Line 384: Isn’t this imputable to the architecture choice more than the type of generative model?  Especially upsampling layers as empirically shown here: https://arxiv.org/pdf/2010.14356 ?

---

> ### Author Response · Authors · 2024-11-22
> **Reply to Reviewer 2uM7 (Part 1)**
>
> We sincerely thank reviewer 2uM7 for the thorough and insightful evaluation of our work. We appreciate the recognition of how our methodological choices help bridge the efficiency gap between diffusion models and GANs without resorting to distillation techniques.
>
> Below we address the concerns mentioned in the review.
>
> **W1: 2 parts is not very clear in its explanations.**
>
> We sincerely thank reviewer 2uM7 for highlighting this issue and providing detailed revision advice. We have updated the paper in accordance with the reviewer's suggestions.
>
> > **"straightness should be defined for a time $s$, as $S(v, s)$"**
>
> Regarding the definition of straightness, it is introduced in [1], on page 7, equation (3). It is defined as the deviation of the sampling trajectory from the straight transport trajectory. The time $t$ is integrated out along the sampling trajectory. We have added a section of code in Appendix A.9.2 for selecting sample time points of equal straightness, which includes the numerical estimation of straightness. We hope this clarifies the definition.
>
> [1] Xingchao Liu, Chengyue Gong, Qiang Liu: Flow Straight and Fast: Learning to Generate and Transfer Data with Rectified Flow. ICLR 2023
>
> **W2: Concerns on dataset consistency for different models**
>
> In response to the concerns about dataset consistency, we have ensured that RFWave is compared to other models trained on consistent datasets. We acknowledge that the original description of the datasets may have led to some confusion, and we have clarified this in the updated paper.
>
> On page 7, lines 331-334, we explain our approach when benchmarking against diffusion vocoders. We trained separate models on the LibriTTS (speech), MTG-Jamendo (music), and Opencpop (vocal) datasets. Each model is tested on its respective dataset to ensure a comprehensive comparison across various audio categories.
>
> When comparing RFWave to widely used GAN-based models, we trained a model on the LibriTTS dataset and evaluated its in-domain performance using the LibriTTS test set. Additionally, to assess the out-of-domain generalization ability, we tested this LibriTTS-trained model on the MUSDB18 test subset.
>
> For discrete EnCodec token inputs, as mentioned in lines 338-341, we followed convention of EnCodec and MBD by training a universal model on a large-scale dataset. This dataset combines Common Voice 7.0 and clean data from DNS Challenge 4 for speech, MTG-Jamendo for music, and FSD50K and AudioSet for environmental sounds. This approach ensures consistency since the same datasets are used by both EnCodec and MBD.
>
> **W3: Comparison with Latent Diffusion Models, such as a fine-tuned version of Stable Audio Open**
>
> LDM is a highly influential work in the text-to-image domain, utilizing diffusion models to predict a compact representation instead of directly predicting the original image, significantly improving computational efficiency.
>
> Similarly, the complex spectrogram is much shorter in length compared to waveform. From this perspective, the complex spectrogram can be considered analogous to the "latent compact representation" in LDM. Therefore, operating on the complex spectrogram level enhances model efficiency.
>
> However, our task scenario differs from Stable Audio Open, which addresses the text-to-audio problem by generating audio effects based on a text prompt. Our task involves reconstructing audio waveforms from a more compact speech feature. In fact, our model can serve as a part of a text-to-audio system; specifically, RFWave has the potential to replace the decoder in Stable Audio Open.
>
> We trained an RFWave model to reconstruct waveforms from Stable Audio Open latents and obtained some preliminary results. The demo audio samples can be found [here](https://rfwave-demo.github.io/rfwave/stable_open_demo_full.html). While our model demonstrates basic functionality, it requires further training to converge fully. Our training setup differs from Stable Audio Open's, whose processed dataset, while sourced from public data, has not been released.

---

> ### Author Response · Authors · 2024-11-22
> **Reply to Reviewer 2uM7 (Part 2)**
>
> **Q2: Clarification Fourier features. How we concatenate the conditional inputs?**
>
> Fourier features were initially proposed in [1] and subsequently introduced to the diffusion model in [2]. These features are high-frequency periodic functions designed to amplify small variations in the subband's noisy sample $X_t^{i_{sb}}$. They are calculated using $\sin(2^n\pi X_t^{i_{sb}})$ and $\cos(2^n\pi X_t^{i_{sb}})$, where $n$ ranges over a set of integers $\{n_{\text{min}}, \ldots, n_{\text{max}}\}$.  Both the Fourier features and conditional inputs are concatenated to $X_t^{i_{sb}}$ along the channel dimension, as they have the same length. In the paper, we have updated Figure 1 and Section 3.1 to make this part clearer.
>
> [1]Matthew Tancik, Pratul P. Srinivasan, Ben Mildenhall, Sara Fridovich-Keil, Nithin Raghavan, Utkarsh Singhal, Ravi Ramamoorthi, Jonathan T. Barron, Ren Ng: Fourier Features Let Networks Learn High Frequency Functions in Low Dimensional Domains. NeurIPS 2020
>
> [2] Diederik P. Kingma, Tim Salimans, Ben Poole, Jonathan Ho: Variational Diffusion Models. CoRR abs/2107.00630 (2021)
>
> **Q3: Line 205: If interleaved then shouldn’t be $2d_s$?**
>
> The previous expression wasn't clear enough. Initially, when we stated that "$d_s$ denotes the dimension of a subband's complex spectrum," we intended to convey that the dimension includes the interleaved real and imaginary parts that form a complex spectrogram (not in the traditional complex data type sense). Thus, $d_s$ encompasses both the real and imaginary dimensions. Thank you for pointing this out. We have clarified the expression by changing "$d_s$ denotes the dimension of a subband's complex spectrum" to "$d_s$ represents the number of frequency bins in a subband's complex spectrum," and updated other instances in the text to $2d_s$ for greater clarity.
>
> **Q4&Q5: Mean-variance normalization in waveform equalization and STFT normalization**
>
> In the time-domain method, PQMF splits the full-band waveform into subbands, followed by mean-variance normalization applied to these subband waveforms (each subband tracks its own mean and variance). The equalized subbands are then merged to form an equalized waveform, which becomes $X_1$ (data) used by the time-domain model.
>
> In the frequency-domain method, the complex spectrogram is extracted from the non-equalized waveform. Dimension-wise mean-variance normalization is performed on the complex spectrogram (each frequency bin tracks its own mean and variance). The output from this normalization process serves as $X_1$ (data) for the frequency-domain model.
>
> The mean-variance normalization employs exponential moving average statistics during training. With these statistics frozen, the normalization can be inverted during inference.
>
> **Q6: Line 246: Why compute the standard deviation and not the energy altogether?**
>
> Figure A.1 demonstrates that energy changes consistently align with the standard deviation. Moreover, mean-variance normalization is more commonly employed in MSE optimization. Therefore, we opt to divide by the standard deviation. Equation 5 can be viewed as the mean-variance normalization of $(X_1 - X_0)$ and $v$, where the mean term is canceled out.
>
> **Q8: Isn’t $X_0$ the data point and $X_1$the noisy point?**
>
> In Rectified Flow [1], $X_0$ represents noise and $X_1$ represents data. In Flow Matching [2], $X_0$ is data and $X_1$ is noise. We adhere to the Rectified Flow notation.
>
> [1] Xingchao Liu, Chengyue Gong, Qiang Liu: Flow Straight and Fast: Learning to Generate and Transfer Data with Rectified Flow. ICLR 2023
>
> [2] Yaron Lipman, Ricky T. Q. Chen, Heli Ben-Hamu, Maximilian Nickel, Matthew Le: Flow Matching for Generative Modeling. ICLR 2023
>
> **Q9: Why snake activation enhances out-of-domain data generation capabilities?**
>
> We adopt the conclusions from BigVGAN [1]. As stated in the BigVGAN paper, in Section 3.2, the last paragraph mentions: "We demonstrate that the proposed Snake-based generator is more robust for out-of-distribution audio samples unseen during training, indicating strong extrapolation capabilities in universal vocoding tasks. See Figure 2 and Appendix D for illustrative examples; BigVGAN-base without filter using snake activations is closer to the ground-truth sample than HiFi-GAN."
>
> [1] Sang-gil Lee, Wei Ping, Boris Ginsburg, Bryan Catanzaro, Sungroh Yoon: BigVGAN: A Universal Neural Vocoder with Large-Scale Training. ICLR 2023

---

> ### Author Response · Authors · 2024-11-22
> **Reply to Reviewer 2uM7 (Part 3)**
>
> **Q10: Line 384: Isn’t this imputable to the architecture choice more than the type of generative model?**
>
> The generator architecture is indeed important, particularly regarding the artifacts introduced by the upsampling layers, as the reviewer pointed out. However, the training criterion and the type of generative model are also crucial factors. For instance, in GAN-based models, the multi-period discriminator in HiFi-GAN [1] and the HingeGAN loss in DAC [2] significantly enhance audio quality.
>
> [1] Jungil Kong, Jaehyeon Kim, Jaekyoung Bae:
> HiFi-GAN: Generative Adversarial Networks for Efficient and High Fidelity Speech Synthesis. NeurIPS 2020
>
> [2] Rithesh Kumar, Prem Seetharaman, Alejandro Luebs, Ishaan Kumar, Kundan Kumar: High-Fidelity Audio Compression with Improved RVQGAN. NeurIPS 2023
>
>
> **Thanks for pointing out the typos - they have been corrected in the revised version.**

---

### Author Response · Authors · 2024-12-03
**Global Response to the Reviews**

Dear Reviewers,

We sincerely thank all reviewers for their thorough evaluation of our manuscript and their insightful feedback.

We are deeply grateful that the reviewers recognized RFWave's significant contributions in bridging the efficiency gap between diffusion and GAN-based waveform reconstruction models while maintaining high quality (2uM7, c6cp, kzjW, rq1Z). Our technical innovations, including parallel subband processing (2uM7, kzjW, rq1Z), novel loss functions (2uM7, 5nT1, kzjW, rq1Z), and specialized sampling strategy (2uM7, rq1Z), have enabled audio generation "much faster than in real-time" (c6cp) with "superior sound quality" (5nT1) while using fewer parameters and less memory than baselines (rq1Z, kzjW).

We are encouraged by the recognition that our introduced techniques will be "highly valuable for future follow-up works on flow models in audio processing" (rq1Z).

---

### Meta-Review · Area_Chair_3t9e · 2024-12-20

**Metareview:**

The paper proposes a generative model for audio signals named RFWave, a diffusion-type solution. Inspired by rectified flow studies, the authors introduce an enhanced sampling strategy and novel loss functions designed to generate high-quality audio signals, addressing challenges such as ensuring that nearly-silent portions of the signal do not produce noisy outputs. RFWave employs a convolutional architecture based on ConvNeXtV2 and outperforms various baseline methods, including GAN and diffusion-based solutions, across a comprehensive set of experiments. Achieving these results with only ten sampling steps makes it a particularly remarkable solution.
The paper was reviewed by five experts, and the consensus among reviewers was to accept the work. The AC agrees with the reviewers' evaluation and recommends acceptance.  **Congratulations** on this excellent work! Please revise the study based on the author-reviewer discussions. Once again, congratulations!

**Additional Comments On Reviewer Discussion:**

A lengthy and constructive discussion took place between the reviewers and the authors, focusing on expanded results, clarifying the contributions, and addressing methodological details (e.g., time vs. frequency domain solutions). These clarifications helped better position the work. Following the discussion, two reviewers increased their scores to eight.  Given the consensus among reviewers on the significance of the work, the AC recommends acceptance. **Congratulations**.

---

### Decision · Program_Chairs · 2025-01-22

Accept (Poster)